# The σ<sup>B</sup> alternative sigma factor circuit modulates noise to generate different types of pulsing dynamics

**Torkel E. Loman, James C. W. Locke** *

Sainsbury Laboratory, University of Cambridge, Cambridge, United Kingdom

* james.locke@slcu.cam.ac.uk

## Abstract

Single-cell approaches are revealing a high degree of heterogeneity, or noise, in gene expression in isogenic bacteria. How gene circuits modulate this noise in gene expression to generate robust output dynamics is unclear. Here we use the *Bacillus subtilis* alternative sigma factor σ<sup>B</sup> as a model system for understanding the role of noise in generating circuit output dynamics. σ<sup>B</sup> controls the general stress response in *B. subtilis* and is activated by a range of energy and environmental stresses. Recent single-cell studies have revealed that the circuit can generate two distinct outputs, stochastic pulsing and a single pulse response, but the conditions under which each response is generated are under debate. We implement a stochastic mathematical model of the σ<sup>B</sup> circuit to investigate this and find that the system's core circuit can generate both response types. This is despite one response (stochastic pulsing) being stochastic in nature, and the other (single response pulse) being deterministic. We demonstrate that the main determinant for whichever response is generated is the degree with which the input pathway activates the core circuit, although the noise properties of the input pathway also biases the system towards one or the other type of output. Thus, our work shows how stochastic modelling can reveal the mechanisms behind non-intuitive gene circuit output dynamics.

**Data Availability Statement:** Scripts for generating all of the figures in this article, as well as the simulations on which they are based, can be found at https://gitlab.developers.cam.ac.uk/slcu/teamjl/loman_locke_2023.

## Author summary

Experimental advances have enabled the measurement of the dynamics of gene regulatory networks at the single-cell level. This has revealed surprising heterogeneity, or noise, in gene expression between genetically identical cells. This noise can be beneficial, for example by allowing a bacterial population to 'bet-hedge' against future environmental change by having a few cells randomly enter a stress prepared state. Here, we use mathematical modelling to investigate a noisy gene regulatory circuit, the σ<sup>B</sup> mediated general stress response of the bacterium *Bacillus subtilis*. By creating a stochastic model of the σ<sup>B</sup> network, we can replicate the two different response behaviours the system has previously been shown to produce. Interestingly, the first of these behaviours (a single response pulse) is non-stochastic in nature, while the other (stochastic pulsing) is distinctly stochastic. We scan system parameters (properties) to determine how these affect which

**Funding:** The work in the Locke laboratory is supported by a fellowship from the Gatsby Foundation (GAT3272/GLC to JCWL). TEL has received funding from the European Union's Horizon 2020 research and innovation programme under the Marie Sklodowska-Curie grant agreement No. 721456. The funders had no role in study design, data collection and analysis, decision to publish, or preparation of the manuscript.

**Competing interests:** The authors have declared that no competing interests exist.

behaviour the system produces. We show that relatively minor perturbations can push the system from a single pulse response to a stochastic pulsing regime, helping explain previous contradictory experimental results. Our work furthers understanding of how noise in gene expression can enable novel gene circuit dynamics.

# 1 Introduction

As we gain more detailed knowledge of the composition of biochemical systems, the use of mathematical chemical reaction network (CRN) models is not only becoming increasingly feasible, but also an important tool to discern how these networks process information. CRN models have been used for such wide applications as the study of animal development [1, 2], plant circadian rhythms [3, 4], and bacterial stress response [5, 6]. In addition, they are an important tool in synthetic biology, where they can be used to design circuits with desired regulatory properties [7]. Traditionally such models have been deterministic in nature. However, stochastic modelling techniques are now often used to capture noisy system dynamics [8]. Such noise can both be caused by low molecule numbers of system components (intrinsic noise) and external quasi-random effects such as cell cycle stage and variability in cellular energy supply (extrinsic noise) [9–11]. Noise in gene expression has been shown to be important to the dynamics of many systems, enabling new and sometimes even beneficial biological phenotypes [12–16]. In this article, we use the general stress response alternative sigma factor of *Bacillus subtilis*, σ$^B$, as a model system to study the influence of noise on the output of a cellular network.

Sigma factors are an essential component of bacterial RNA polymerase, allowing it to recognise and bind to genomic promoter regions. Under standard environmental conditions, a so-called housekeeping sigma factor is used. However, in response to environmental cues, an alternative sigma factor can be activated, replacing the housekeeping sigma factor in RNA polymerase and redirecting the bacterium's transcriptional program [17–21]. σ$^B$ of *B. subtilis* is one of the most well-studied alternative sigma factors. It responds to two classes of stress: energy stress (caused by ATP depletion) and environmental stress (caused by factors such as ethanol, heat, or salt). In response to these stresses, it activates approximately 200 genes involved in the general stress response [22–24]. σ$^B$ activity is controlled by a core circuit, which can be activated by two distinct upstream pathways (which are triggered by energy and environmental stress, respectively) (Fig 1) [22, 24, 25].

The core σ$^B$ circuit consists of σ$^B$, its anti-sigma factor (RsbW), and its anti-anti-sigma factor (RsbV). In the absence of stress, σ$^B$ is bound to, and kept inactive by, RsbW. Meanwhile, RsbV is kept inactivated by a phosphate group. The energy and environmental stress sensing pathways each activate a phosphatase (RsbP and RsbTU, respectively) [26, 27]. These dephosphorylate RsbV, activating it. Active RsbV binds RsbW in a partner switching mechanism that also releases (and thus activates) σ$^B$ [28–31]. Active σ$^B$ will not only activate the general stress response, but also activate an operon containing itself, RsbW, and RsbV, creating a mixed positive/negative feedback loop [32, 33]. RsbW also acts as a kinase, rephosphorylating RsbV, thus resetting the system in the absence of stress [30].

Advances in single-cell fluorescent microscopy have enabled the study of dynamic gene expression at the single-cell level [34, 35]. These techniques have been deployed on alternative sigma factor systems, with reporters for alternative sigma factor activity often displaying heterogeneous activation dynamics (even across isogenic populations experiencing homogeneous inputs) [36–42]. It has been suggested that this heterogeneity can be generated by the

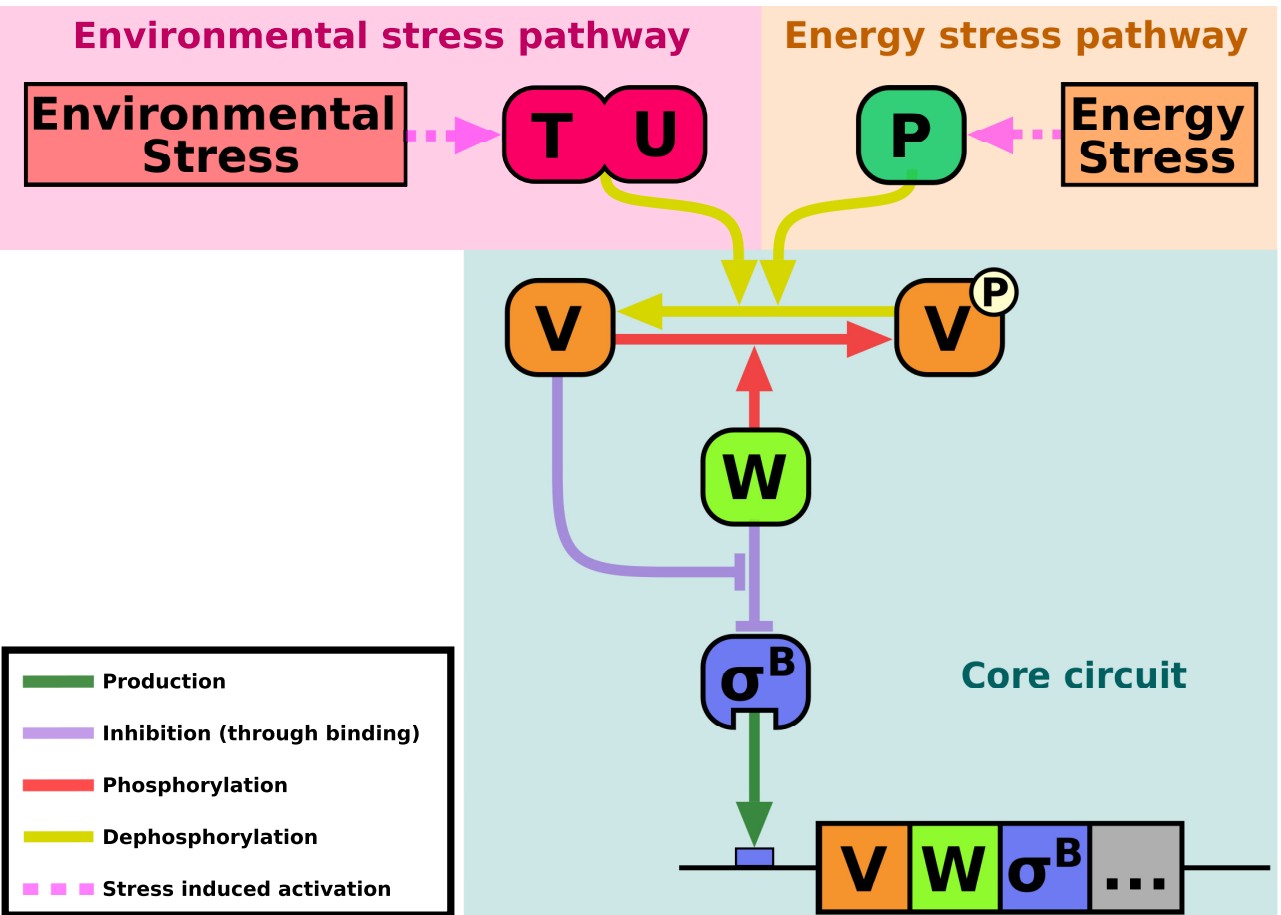

**Fig 1. The σ^B regulatory network consists of a core circuit, which is activated by two distinct upstream pathways.** Under non-stress conditions, σ^B is bound, and held inactive by, its anti-sigma factor (RsbW, W in the figure). An anti-anti-sigma factor (RsbV, V in the figure) is inactivated by a phosphate group. The upstream pathways are triggered by two distinct types of stress (environmental and energy stress), each activating their respective phosphatase (RsbTU and RsbP, respectively). These will dephosphorylate, and thus activate, RsbV. Once activated, RsbV binds RsbW, which simultaneously releases σ^B in a partner switching mechanism. This permits σ^B to activate the general stress response of *B. subtilis*. RsbW is a kinase that re-phosphorylates RsbV, allowing RsbW to re-bind σ^B and shut the activation off. In addition, σ^B activates the production of itself, RsbW, and RsbV, creating a mixed positive/negative feedback loop. The environmental stress response phosphatase complex, RsbTU, consists of the co-factor RsbT and the phosphatase RsbU (T and U, respectively, in the figure). The availability of RsbT is controlled by a multi-protein complex called the stressosome and the phosphatase RsbX (both not depicted in the figure for simplicity). The energy stress response phosphatase, RsbP (P in the figure), also depends on a second protein, RsbQ (not depicted in this figure).

amplification of intrinsic cellular noise, and that it may create beneficial, variable, phenotypes through bet-hedging behaviours [43]. Investigations of σ^B using single-cell fluorescent micros- copy have shown that the two stress types (energy and environmental stress) can produce two distinct response behaviours. In response to energy stress, σ^B displays a persistent stochastic pulsing behaviour. It has been observed that σ^B is activated in short (∼ 1 hour) activity pulses, randomly and independently distributed across the population and time [36, 38]. The environ- mental stress response is instead a single response pulse, where the population displays a syn- chronised pulse (∼ 1 hour long) at the time of stress onset, but the system thereafter remains OFF as stress persists [44]. A model of a simplified σ^B network was implemented in [36]. Later, a model based on the full network CRN was implemented in [45]. Using this model, the authors showed how the relative stoichiometry of the synthesis rates of σ^B, RsbW, and RsbV

can cause the network to function as an ultrasensitive negative feedback loop, generating pulses. The model can explain the two pathways' different responses by assuming that environmental stress produces a tightly controlled supply of phosphatase (triggering the core circuit on stress onset only), while energy stress instead produces a naturally fluctuating phosphatase supply (repeatedly triggering the core circuit).

Further experiments, however, have demonstrated that the two response behaviours are not unique to their respective pathways. In [37] it was shown that energy stress could produce a single response pulse, without following stochastic pulses. In addition, it was shown that the environmental stress response, in backgrounds where the stressosome (a multi-protein complex that controls the availability of the RsbT component of RsbTU) was mutated, could also produce a stochastic pulsing-type behaviour. The system's ability to generate both behaviours from both pathways suggests that it is the core $\sigma^B$ circuit that is able to produce both a single response pulse and stochastic pulsing. It is possible that minor perturbations in the two stress sensing pathways could bias the core circuit towards either behaviour.

In this article, we develop a stochastic model of the $\sigma^B$ network, based on a previous deterministic model of $\sigma^B$ regulation [45] (hereafter referred to as the Narula model). By considering noise in all system components we confirm that the core circuit can generate both stochastic pulsing and single pulse dynamics without requiring different assumptions about the noise properties of upstream energy and environmental stress pathways. Next, we find a minimal set of parameters that tune the system's response behaviour, and investigate how the tuning of these parameters modulates which behaviour is displayed. We show how a single system parameter (the effective dephosphorylation rate of RsbV) is the primary determinant of which behaviour is produced. Furthermore, we show how the system transitions from a single pulse response, to stochastic pulsing, to oscillations as this parameter is increased. This provides an explanation for recent data showing that the energy stress pathway and the environmental stress pathway can display both stochastic pulsing and single pulse response dynamics. Finally, we demonstrate how properties of noise in the system's upstream pathways may still bias it towards either response behaviour.

## 2 Results

### 2.1 The core $\sigma^B$ circuit can produce both the stochastic pulsing and the single response pulse behaviour

The Narula model recreates the energy and environmental stress response behaviours (stochastic pulsing and a single response pulse, respectively) by assuming that the two upstream pathways present qualitatively different input processes to the core $\sigma^B$ circuit [45]. Since then, it has been shown experimentally that both pathways can generate both responses under specific environmental or genetic perturbations [37]. This suggests that the core circuit can generate both behaviours. The Narula model relies on a stochastic input process (to a deterministic model) to generate noisy behaviours. By instead using a stochastic CRN interpretation of this model, we investigated the core circuit's ability to generate both response behaviours from constant stress inputs.

To implement a stochastic CRN, we used the chemical Langevin equation (CLE) [46]. The CLE, which is an established technique for modelling noise in gene regulatory networks [47–50], is an approximation of the Gillespie algorithm [51, 52]. The Gillespie algorithm permits accurate simulations of the actual reaction events of a CRN, correctly taking into account the inherent randomness of the reactions to create stochastic simulations. Gillespie simulations, however, have long simulation runtimes. By approximating the system as a stochastic differential equation (SDE), the CLE permits fast stochastic simulations. While the

exactness of this approximation is reduced with increasing noise levels, it is more accurate than other approximations such as the linear noise approximation [53]. For our application, the faster simulation times the CLE permits were important for performing the parameter sweeps carried out later in the paper. The CLE also allows independent tuning of the system's noise amplitude through the introduction of a system size parameter $\Omega$. Using an equivalent formulation, our noise term in the CLE contains a noise scaling parameter (where $\eta = 0$ means no noise and $\eta = 1$ means unscaled noise), and we use this to investigate the noise amplitude's effect on the system. To enforce non-negativity in our CLE simulations, we used absolute values of any negative numbers in the noise terms [54]. To confirm that our conclusions are not due to the assumptions of the CLE, throughout the paper we verify that key results can be reproduced using the Gillespie algorithm [51, 52]. We note that the CLE and Gillespie algorithm both model intrinsic noise only, and not extrinsic noise. While sources of extrinsic noise (such as cell cycle variations) could be added to the model [55], we omitted these as current experimental evidence points to transcriptional noise driving alternative sigma factor pulse initiation and not extrinsic factors such as the cell cycle [36, 38].

We first simulated our CLE adaptation of the Narula model under a range of conditions to test what output dynamics are generated. By subjecting the model, at various noise levels, to various degrees of stress, we noted that modulation of $\eta$ (the degree of noise) tunes the amplitude of the transient response pulse (Fig 2A and 2B). However, while such modulations created increased fluctuations in the asymptotic state, they were unable to produce stochastic pulsing (S1 Fig). It is known that pulses are related to oscillations [56] (a phenomenon that can also be further emphasised by noise [57, 58]). By screening bifurcation diagrams of all parameters we identified $k_{K2}$ (the re-phosphorylation rate of the anti-anti-sigma factor RsbV) as the only parameter that can induce oscillations under non-absurd tunings (S2 and S3 Figs). Tuning of a combination of parameters can also induce pulsing. However, we wished to minimise the number of model parameter values we modified. By modifying a smaller set of parameters we reduce the dimensionality of the parameter space we need to explore, reducing computational complexity and enabling more complete scans. We thus selected $k_{K2}$ only as our proxy for the system's proneness to pulsing.

We wished to determine to what degree the system can generate the single response pulse and stochastic pulsing behaviours. To do this, we developed an automatic measure that takes a parameter set as input, and returns the magnitude of both behaviours (Fig 2C and 2D). An extensive description of the measure can be found in Section 4.5.1, while this paragraph contains a briefer description. First; we utilise that $\sigma^B$ pulse durations rarely exceed 1 hour to divide $\sigma^B$ stress response simulations (with stress added at time $t = 0$) into two different time phases: a transient phase ($0 - 5$ hours after stress addition) and an asymptotic phase ($> 5$ hours after stress addition). Next, we find the maximum $\sigma^B$ activity in the transient and asymptotic phases, as well as the mean $\sigma^B$ activity in the asymptotic phase. The degree of the single response pulse behaviour, in a single simulation, is measured as the maximum $\sigma^B$ activity in the transient phase divided by the maximum activity in the asymptotic phase. Similarly, for stochastic pulsing, the definition is the maximum activity in the asymptotic state divided by the mean activity in the asymptotic state. In both cases, to gain a more precise measure for a given parameter set, we take each behaviour's mean magnitude across a large number of simulations (between 50 and 200).

Using these measures, we evaluated the system's proneness to both behaviours across $\eta - k_{K2}$ space (Fig 2E and 2F). This revealed how, in addition to the single pulse response behaviour, as intrinsic noise is added to the ODE model (by increasing $\eta$), the stochastic pulsing behaviour emerges (but only when $k_{K2}$ is small enough). Through sample

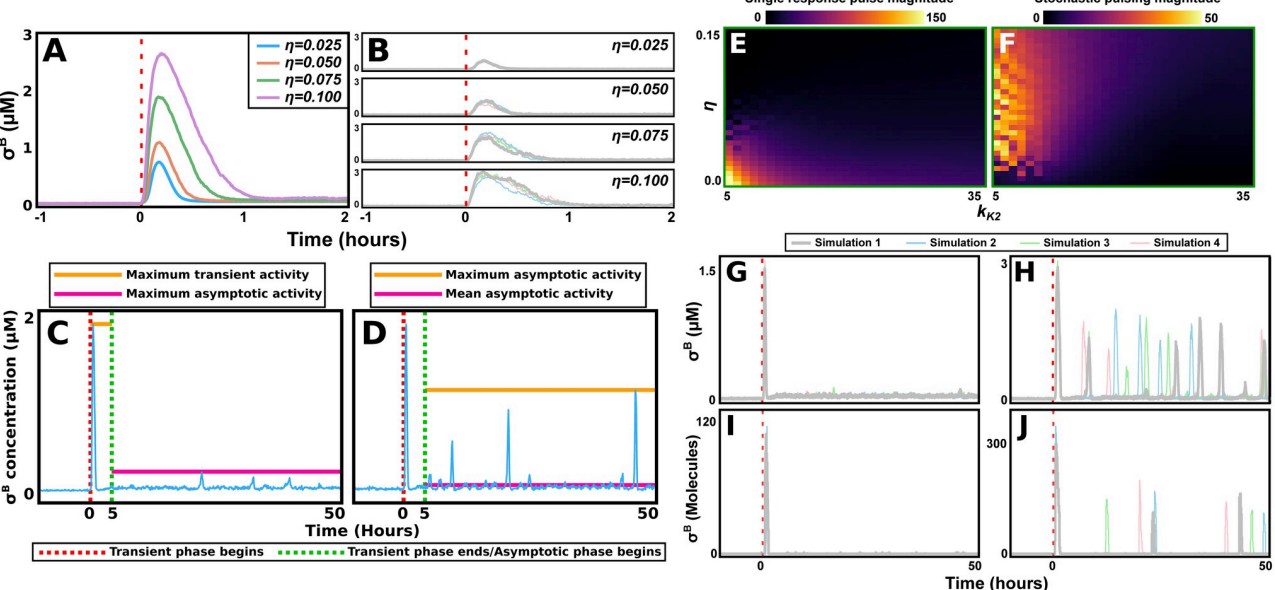

**Fig 2. The core σ^B circuit is capable of generating both response behaviours.** (A,B) For various noise amplitudes ($\eta$), the CLE adaptation of the Narula model is exposed to a stress step (at $t = 0$, red dashed line). (A) Mean [σ^B] in response to the input (average over $n = 150$ simulations). The amplitude of the single response pulse increases with noise. (B) For each value of $\eta$, four different simulations are shown. There is little variation between the individual trajectories. (C,D) Illustration of our measures for the degree with which the system exhibits the single response pulse (C) and stochastic pulsing (D) behaviours. A simulation is divided into a *transient phase* ($t \in [0.0, 5.0]$) and an *asymptotic phase* ($t \in [5.0, 200.0]$, but we note that in this figure the x-axis is cut to $t = 50$ to better display both phases). Next, we find *the maximum activity in the transient phase*, *the maximum activity in the asymptotic phase*, and *the mean activity in the asymptotic phase*. The single response pulse measure (C) is defined as the maximum transient activity (orange line) divided by the maximum asymptotic activity (magenta line). The stochastic pulsing measure (D) is defined as the maximum asymptotic activity (orange line) divided by the mean asymptotic activity (magenta line). In practice, a mean measure over several ($n > 50$) simulations is always used. (E,F) The parameters $\eta$ (noise amplitude) and $k_{K2}$ (a proxy for the system's proneness to oscillations) are varied. For each parameter combination, the maximum magnitude of the single response pulse (E) and stochastic pulsing (F) behaviours that can be achieved by varying the parameter $p_{stress}$ (20.0 μM< $p_{stress}$ < 200.0 μM) is found and plotted. (G) The single response pulse behaviour is maximised at ($k_{K2}, \eta$) = (15.0$hr^{-1}$, 0.04). For these values, four simulations are shown. (H) The stochastic pulsing behaviour is maximised at ($k_{K2}, \eta$) = (9.0$hr^{-1}$, 0.06). For these values, four simulations are shown. (G,H) These simulations demonstrate that the CRN of the Narula model can generate both behaviours while exposed to intrinsic noise only. (I,J) It is possible to recreate both response behaviours, single pulse response (I) and stochastic pulsing (J), using the Gillespie algorithm. This demonstrates that the responses are not dependent on the modelling approach used. Parameter values and other details on simulation conditions for this figure are described in S1–S3 Tables.

simulations, we demonstrated that the core circuit can generate both response behaviours, and that this holds even when the phosphatase concentration in the upstream pathways remains constant (Fig 2G and 2H, and S4 Fig). We also confirmed that these behaviours can be recreated using the Gillespie algorithm (Fig 2I and 2J). For the Gillespie algorithm simulations, as we could not tune a noise parameter $\eta$, instead we modified a larger set of parameters to generate the different dynamics (S3 Table). Finally, we investigated the initiation of the stochastic pulses, noting that these typically are preceded by a drop in RsbW concentration (S5 Fig).

## 2.2 The primary determinant of the system's response behaviour is the total RsbV dephosphorylation rate

We have shown that through modulation of $\eta$ and $k_{K2}$, the system can reproduce both stochastic pulsing and single response pulse behaviours. However, $\eta$ and $k_{K2}$ are properties of the core circuit, and these are identical for the energy and environmental stress responses. Thus, the system must be able to generate both a single response pulse and stochastic pulsing while

keeping $\eta$ and $k_{K2}$ fixed and varying the properties of the upstream pathway only. In the Narula model, this pathway is determined by the parameters $p_{stress}$, $k_{B5}$, $k_{D5}$, and $k_P$. These denote the total amount of phosphatase ($p_{stress}$), and the rates at which it binds ($k_{B5}$), dissociates from ($k_{D5}$), and dephosphorylates RsbV ($k_P$), according to these reactions:

$$P + RsbV^P \underset{k_{B5}}{\overset{k_{D5}}{\longleftarrow}} P{-}RsbV^P \overset{k_P}{\longrightarrow} P + RsbV$$

Here, $k_{B5}$, $k_{D5}$, and $k_P$, are fixed throughout a simulation, while $p_{stress}$ denotes the post-stress amount of phosphatase (with the phosphatase amount changed from $p_{init}$ to $p_{stress}$ at the time of stress onset).

We first wanted to determine for which set of parameters the core circuit can generate both response behaviours as the upstream parameters alone are varied. To do this, we defined a measure of the system's ability to robustly generate both response behaviours (stochastic pulsing and single response pulse) while varying only the upstream parameters (S6 Fig). By scanning this measure across $\eta - k_{K2}$ space, we found that $(\eta, k_{K2}) = (0.025, 7.0 hr^{-1})$ optimises this measure (S7 Fig), meaning that this parameter combination allows the core circuit to generate both behaviours, with the behaviour that is generated being dependent on the values of upstream parameters (S8 Fig). We next fixed $\eta$ and $k_{K2}$ at this optimum, and then proceeded to investigate how the parameters $p_{stress}$, $k_{B5}$, $k_{D5}$, and $k_P$ modulate the system's response.

To determine which parameters governing the upstream pathways are most important for determining the response behaviours, we next scanned the two behaviours' magnitudes across $p_{stress}$-$k_{B5}$-$k_{D5}$-$k_P$ space (S9 Fig). Our scans show that both behaviours occur in regions along the curve $p_{stress} \cdot k_P = C$, for various values of $C$ (Fig 3A and 3B). This suggests that the total RsbV dephosphorylation efficiency (the amount of phosphatase, $p_{stress}$, times its dephosphorylation efficiency, $k_P$) is an important determinant for the system's response. We next performed parameter substitutions $p_{prod} = p_{stress} \cdot k_P$ ($p_{prod}$ = total RsbV dephosphorylation efficiency) and $p_{frac} = p_{stress}/k_P$. By redoing the magnitude scans using this substitution, we confirmed that the total dephosphorylation efficiency, $p_{prod}$, is important for determining the response behaviour (Fig 3C and 3D, and S10–S15 Figs). Finally, we evaluated the behavioural magnitudes' sensitivity to change in the four parameters $p_{prod}$, $p_{frac}$, $k_{B5}$, and $k_{D5}$ (Fig 3E and 3F). This showed that both behaviours are robust as the values of $p_{frac}$, $k_{B5}$, and $k_{D5}$ change, but are sensitive to the value of $p_{prod}$. This suggested that the dynamics encoded by the upstream pathway are primarily defined by the value of $p_{prod}$. Thus, for further analysis, we set $k_{B5} = 3600 \mu M^{-1} hr^{-1}$, $k_{D5} = 18 hr^{-1}$ (their original values) and $p_{frac} = 100 Mhr^1$ (an intermediate value), and let the upstream pathway be defined by the value of $p_{prod}$ only (greatly reducing the dimensionality of the parameter space we need to consider for further analysis).

Next, we investigated how the system's response is modulated by the upstream pathway's critical parameter $p_{prod}$ (the total RsbV dephosphorylation efficiency). We simulated the system for $p_{prod}$ values ranging from small (where the response is absent) to large (where the response saturates) (as $p_{prod} = p_{stress} \cdot k_P$, this increase corresponds to an increase in stress input) (Fig 4). For small values of $p_{prod}$ the system exhibits no response and σ<sup>B</sup> concentrations are low as stress is added. As $p_{prod}$ is increased, the system exhibits the single response pulse behaviour. Here, the amplitude of the pulse increases with the stress (Fig 4D and 4E). This is in agreement with previous experiments claiming the σ<sup>B</sup> environmental stress response to be amplitude modulated [44]. For larger values of $p_{prod}$, the system displays stochastic pulsing (Fig 4F and 4G). The model predicts that the pulse frequency increases with the stress, again similar to experimental data for the energy stress response [36]. Next, the

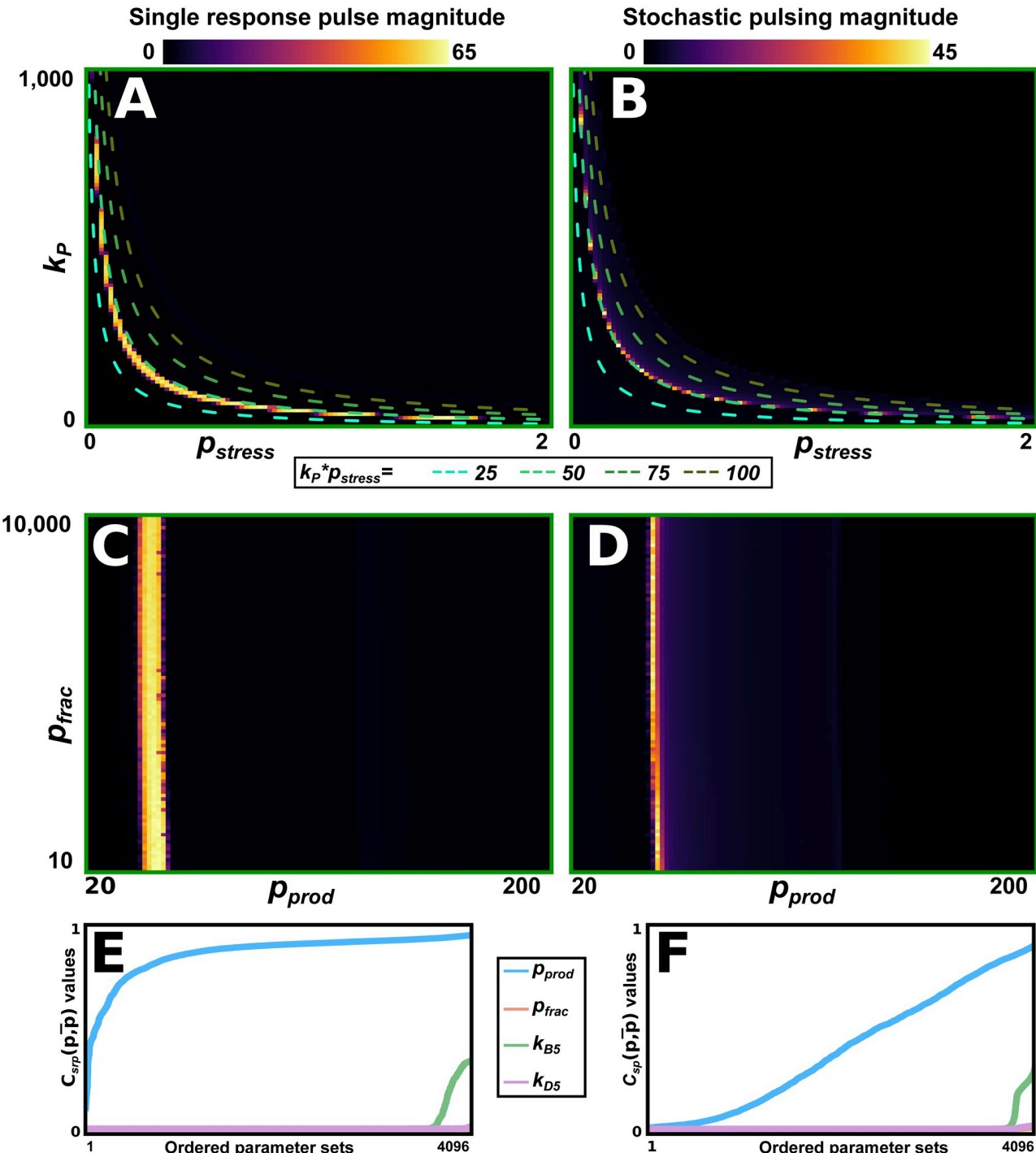

**Fig 3. The effective dephosphorylation rate is the main determinant of which behaviour is produced.** (A,B) The magnitude of the single pulse response (A) and stochastic pulsing (B) behaviours across $k_P$-$p_{stress}$-space. The regions where either behaviour occurs are similar to that of the $k_P \cdot p_{stress}$ = C curve (show for various values, green dashed lines). (C,D) Parameter substitutions generate a parameter $p_{prod} = p_{stress} \cdot k_P$, to which both behaviours are sensitive, and a parameter $p_{frac} = p_{stress}/k_P$, to which both behaviours are insensitive. While the regions corresponding to either behaviour are adjacent, they do not overlap. The stochastic pulsing behaviour exists for slightly larger values of $p_{prod}$, as compared to the single response pulse behaviour. (E,F) For 4096 different parameter sets, we characterise both behaviours' sensitivity to change ($C_{srp}(p, \bar{p})$, E, and $C_{sp}(p, \bar{p})$, F) in the parameters $p_{prod}$, $p_{frac}$, $k_{B5}$, and $k_{D5}$ (Section 4.5.4). We do this by evaluating $C_{srp}(p, \bar{p})$ and $C_{sp}(p, \bar{p})$ for the four different parameters across all 4096 parameter sets. We then put each set of 4096 evaluations in ascending order and plot them in E and F. For a few parameter sets, the behaviours show some sensitivity to $k_{B5}$. However, $p_{prod}$ has the far largest effect on either behaviour. In both cases, changes to $p_{frac}$ and $k_{D5}$ have little effect on the

system. Hence, these lines coincide (both following the x-axis closely). Parameter values and other details on simulation conditions for this figure are described in S2 and S5 Tables.

system transitions into a region of oscillation (Fig 4H–4J). For strong stresses, the system exhibits a single pulse, followed by an elevated level of σ^B activity. The activity in this asymptotic state increases with the magnitude of the stress, however, this increase saturates for large enough stresses (Fig 4K–4M). These last two behaviours have not yet been observed experimentally. A bifurcation-stability analysis of the system revealed that oscillatory behaviours appear for lower values of $p_{prod}$ in stochastic simulations compared to deterministic ones (S16 Fig). This further demonstrates how noise can influence σ^B dynamics. Finally, we demonstrate that a similar transition between single pulse, stochastic pulsing, and oscillatory dynamics can be recreated using the Gillespie algorithm (S17 Fig). Since Gillespie simulations cannot be performed using the substituted parameter $p_{prod}$, we instead used $p_{stress}$ (however, since $p_{prod}$ scales linearly with $p_{stress}$ their transitions are equivalent).

## 2.3 Properties of noise in the upstream pathway can bias the network towards either response behaviour

Next, we investigated how noise in the upstream pathway affects the system's response. We noted that relatively minor perturbations in the stress magnitude can push the system between

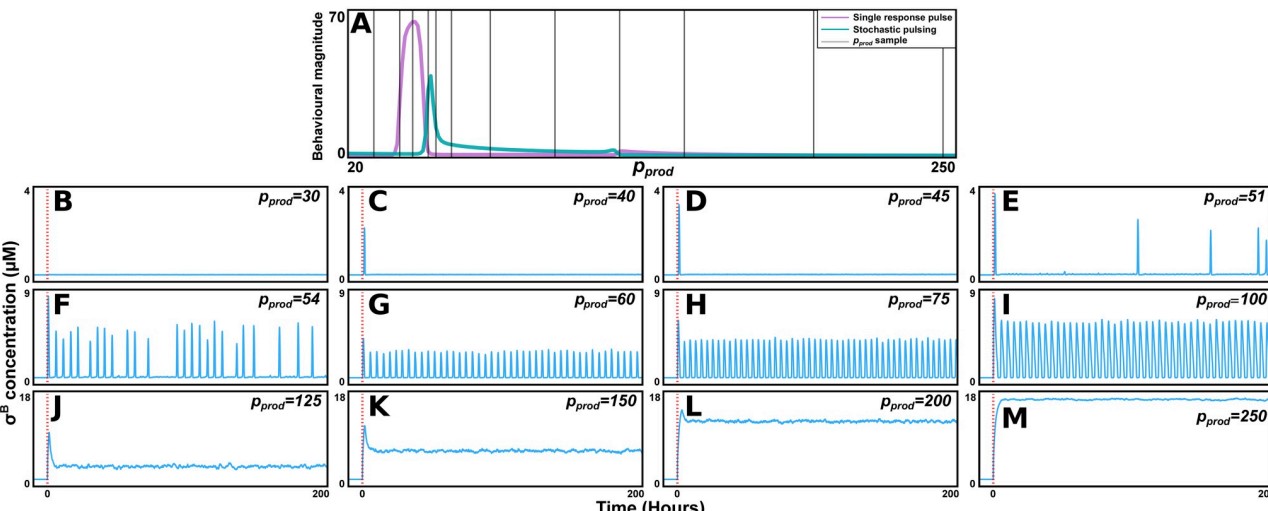

**Fig 4. The system transitions through a range of behaviours as the effective dephosphorylation rate is varied.** (A) The magnitude of the two behaviours for our selected parameter set ($k_{K2}$, $\eta$, $p_{frac}$, $k_{B5}$, $k_{D5}$) = (7.0 h^-1, 0.025, 100.0 μM h^-1, 3600.0 μM^-1h^-1, 18.0 h^-1). 12 different selected values of $p_{prod}$ (used in B-M) are marked with grey lines. (B-M) For 12 different values of $p_{prod}$ a single simulation is displayed (stress added at $t$ = 0, red dashed line). (B) For $p_{prod}$ small, the system does not respond. (C,D) As $p_{prod}$ is increased, the system exhibits a single response pulse. The amplitude increase with $p_{prod}$. (E,F) For larger values of $p_{prod}$, stochastic pulsing is exhibited. The frequency of the pulses increases with $p_{prod}$. (G-I) As the stress is increased further, the system enters a limit cycle. (J-L) For large $p_{prod}$, the system exhibits a single response pulse, and then enters a persistent state of elevated σ^B activity. The activity in this state increases with $p_{prod}$. (M) For large enough stresses, the system saturates at some maximum activity. An expanded version of this figure, including bifurcation analysis of system steady state properties, can be found in S16 Fig. A similar transition, but generated through Gillespie algorithm simulations, can be found in S17 Fig. Parameter values and other details on simulation conditions for this figure are described in S4 and S5 Tables.

the two behaviours (Fig 4E and 4F). However, experiments have shown that the two response types are robust under large variations in stress levels. It is possible that the structure of the upstream pathways makes their respective response behaviours more stable. Previously we have assumed an absence of noise in the availability of the phosphatase. By letting the phosphatase switch between an active and an inactive state, we added intrinsic upstream noise through the CLE. This introduced the parameters $\eta_{amp}$ and $\eta_{freq}$, denoting the amplitude and the frequency of the upstream noise, respectively (S18 Fig and Section 4.2). We next proceeded to investigate how the values of $\eta_{amp}$ and $\eta_{freq}$ affect the system's ability to robustly generate the two behaviours as $p_{prod}$ is varied.

We developed a measure for the system's ability to generate one behaviour distinctly under perturbations to the upstream pathway (with these perturbations previously simplified to modifying the parameter $p_{prod}$) (Section 4.5.3). The measure tests, as $p_{prod}$ is varied, if the system generates one behaviour uniquely while being unable to produce the other one. We scanned this distinctness-of-behaviour measure (for both behaviours) across $\eta_{amp}$-$\eta_{freq}$ space (Fig 5A and 5B). Our scan suggested that the single response pulse behaviour is distinct for

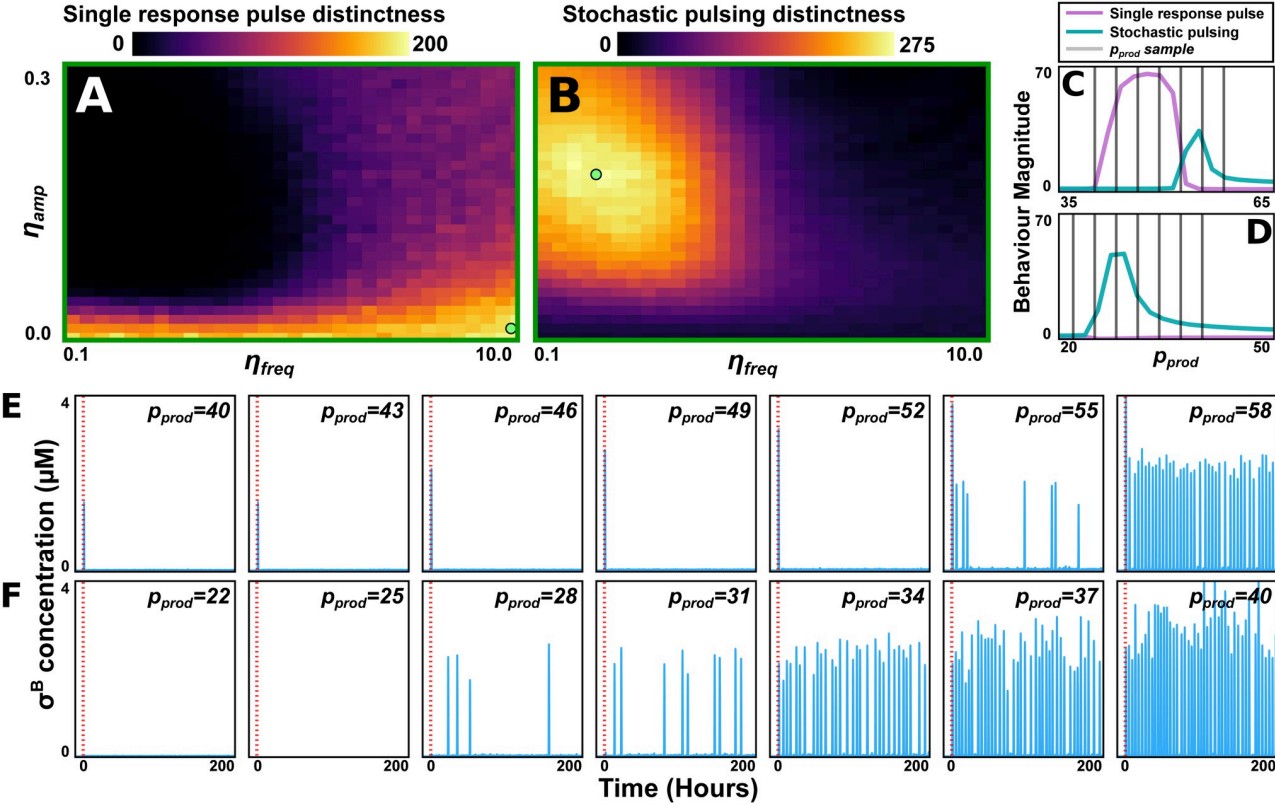

**Fig 5. The properties of the noise in the upstream pathways may bias the system towards either response behaviour.** (A,B) The amplitude ($\eta_{amp}$) and frequency ($\eta_{freq}$) of the upstream pathway's noise is varied. Plots show, for each parameter combination, the distinctness of the single response pulse (A) and stochastic pulsing (B) behaviours. The distinctness (of either behaviour) designates the system's ability to uniquely generate that behaviour (and not the other behaviour) as a parameter is varied (here $p_{prod}$). This measure is described in detail in Section 4.5.3 (there designated $D_{srp}$ or $D_{sp}$). Light green dots mark parameter sets optimising either behaviour's distinctness. (C,D) For the parameter sets that maximise the distinctness of the single response pulse (C) and stochastic pulsing (D) behaviours, the magnitudes of the two behaviours are shown as functions of $p_{prod}$. For each parameter set, 7 selected values of $p_{prod}$ are marked with grey lines. (E,F) For the 7 parameter sets marked in C (E), and D (F), a single simulation is displayed (stress added at $t = 0$, red dashed line). As $p_{prod}$ is varied, the two behaviours are generated much more robustly than what they were for the parameter set in Fig 4. Parameter values and other details on simulation conditions for this figure are described in S6 and S7 Tables.

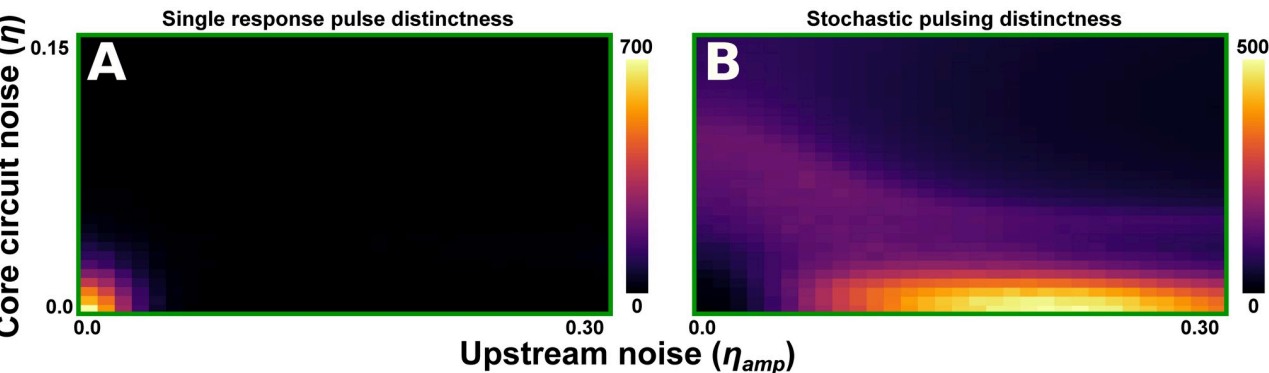

**Fig 6. Core circuit noise compared to upstream noise.** (A,B) For each combination of core circuit noise amplitude ($\eta$) and upstream pathway noise amplitude ($\eta_{amp}$) we vary the parameter $p_{prod}$ (over the interval $20 < p_{prod} < 200$) and calculate each behaviour's distinctness (Section 4.5.3). The total amount of noise in the system, rather than how it is distributed between the core and upstream pathway, is an important determinant for both behaviours' occurrence. (A) The single pulse response behaviour is distinct when both types of noise are low. (B) The stochastic pulsing behaviours require some amount of noise (in either pathway), but do diminish if the total amount of noise becomes too large. It is especially prominent when core circuit noise amplitude ($\eta$) is small and upstream pathway noise amplitude ($\eta_{amp}$) is intermediately valued. Parameter values and other details on simulation conditions for this figure are described in S7 Table.

small values of $\eta_{amp}$. Meanwhile, stochastic pulsing requires low $\eta_{freq}$ and high $\eta_{amp}$. This suggested that while upstream noise is not required to produce either response behaviour, the presence (or lack) thereof, makes either behaviour more robust (Fig 5C–5F).

Finally, our separation of the upstream noise from that of the core circuit allowed us to investigate the relative effect of the two types of noise on the system's behaviour. We measured the relative distinctness of the two behaviours (Section 4.5.3) across $\eta_{amp}$-$\eta$ space, where $\eta$ denotes the amplitude of the core circuit's noise (to reduce combinatorial complexity, we set $\eta_{freq} = 1$). Here, the two stress sensing pathways will have the same core noise amplitude, but different upstream noise amplitudes ($\eta_{amp}$). We note that a network where the core noise is low, and the noise amplitude is low in one pathway and high in the other, should be able to robustly generate the two types of behaviours (Fig 6).

## 3 Discussion

In recent years, the importance of cellular noise to generate biological phenotypes has been demonstrated in a range of systems. These advances have been aided by the development of techniques for dynamic single-cell measurements of cellular components. Such techniques have revealed that the σ^B network responds to two types of inputs (energy and environmental stress) through two distinct response behaviours (a single response pulse and stochastic pulsing, respectively). Here, we use the CLE to implement noise in the Narula model (which was previously based on the deterministic reaction rate equation). This has allowed us to study how noise modulates the network's two responses. By modelling the intrinsic noise of the system's biochemical reactions, we demonstrated how the core circuit can reproduce both response behaviours, without making assumptions about the distinguishing properties of the two upstream pathways. This is in agreement with recent experiments that suggest that both stress types can generate both responses, as experimental conditions are varied [36–38, 44].

To determine how the two upstream pathways generate distinct responses while feeding into a common core circuit, we characterised the parameters defining the pathways.

We showed that the response was mainly determined by the rate of RsbV dephosphorylation ($p_{prod}$). The two responses can thus be explained by the energy stress sensing pathway dephosphorylating RsbV at a higher rate, as compared to the environmental one. Furthermore, we showed how the system transitioned through a range of response behaviours, including single pulse and stochastic pulsing dynamics, as this crucial property was varied.

We also investigated how noise in the upstream pathways affected which behaviour was favoured. We found that the single response pulse behaviour was favoured by low amplitude noise, while the stochastic pulsing behaviour required larger (but not too large) amplitude noise. Furthermore, we noted that the stochastic pulsing behaviour was deterred by high-frequency noise (when fluctuations in the upstream pathway grow larger than the time scale on which the core circuit operates, the core is likely to only see a deterministic mean input). By modelling the amplitude of noise in the core and the pathway separately, we showed how a system with low noise in the core could optimally generate the two response behaviours from two upstream pathways. Here, the environmental stress sensing pathway would be distinctly less noisy than the energy stress sensing pathway. Our unbiased approach thus helps justify assumptions in previous models, where the energy stress pathway was simulated with a gamma distributed Ornstein-Uhlenbeck process whilst the environmental stress was modelled without noise [36, 45], but goes further as it allows us to quantify the effects of different noise levels on each behaviour. Our results suggest that the components of the environmental stress sensing pathway may regulate cellular noise. Indeed, this pathway contains the stressosome protein complex, which controls the availability of phosphatase, and mutations of which permit environmental stress to also generate a stochastic pulsing type response [37, 42]. This suggests that the stressosome complex may reduce cellular noise.

Previous experimental work disagrees on the type of dynamics generated by the energy and environmental stress pathways [36–38, 44]. By showing that the two behaviours are highly related, we can help explain these differing results. We have shown how relatively minor changes in core circuit activation strength, or pathway noise levels, can cause a switch between the two behaviours. This suggests that differences in the stressors used, or possibly the experimental set-up, could affect which response was generated in each set of experiments.

Finally, our results further demonstrate how noise may have non-intuitive effects on dynamic systems [56–58]. This includes our result that the amplitude of the initial response pulse strictly increases with the noise amplitude (Fig 2A and 2B). It also includes our bifurcation-stability analysis of the system (S16 Fig), which demonstrates that oscillatory behaviours appear for lower values of $p_{prod}$ in the stochastic system (as compared to the deterministic one).

To make them tractable for analysis, models must simplify the real system. In this work, we used the chemical Langevin approach (CLE) to model system noise. This is an approximation of the Gillespie algorithm that, due to its simulation speed, allowed us to simulate over a large range of parameters and noise values. However, it is satisfying that we could confirm, using Gillespie algorithm simulations, the key results of our work. This is that the core circuit can generate both single pulse and stochastic pulsing dynamics, and that the system transitions from a single pulse, to stochastic pulsing, to oscillations as the rate of dephosphorylation of RbsV is increased. The fact that simulations using both the CLE and the Gillespie algorithm give qualitatively similar dynamics increases the robustness of our conclusions.

In the future, it will be interesting to examine further, both experimentally and in models, the source of the noise in the σ$^B$circuit. Although technically challenging due to the small size

of the proteins involved and difficulties with fluorescent tags affecting function, it would be powerful to examine the levels of RsbW, RsbV and σ^Bsimultaneously in single cells. This would allow us to use the experimentally measured noise level of each component to directly inform our models. On the modelling side, a more detailed model could include noise due to transcriptional and translational bursts [59, 60], whereas here transcription and translation are combined for simplicity. In addition, we assume uncorrelated noise for each reaction channel, but there is interesting theoretical work that suggests that the coupled expression of genes in operons could lead to correlations in noise that can affect the system dynamics [61]. Further simplifications include that dilution terms are modelled as independent, while in reality they are correlated across all species, as well as the fact that we choose to model intrinsic noise only. Although focusing on intrinsic noise is motivated by our experimental understanding of the system, it would be interesting to examine how extrinsic noise might also affect output dynamics. For example, noise due to the partitioning of molecules at division events is not considered, meaning we could be underestimating the noise due to division [62]. It will be important to address how such additional complexity can affect the σ^B circuit in future models. More generally, stochastic pulsing dynamics have been observed in multiple alternative sigma circuits [38], as well as other stress pathways in microbes [63], and stochastic modelling could allow us to further address whether there are any general principles behind these dynamics. Finally, in Narula (2016) it was shown that the dynamics of the σ^B circuit are robust to competition from other sigma factors [45]. It would be interesting to investigate the effect of intrinsic noise on such competition dynamics.

As methods for stochastic CRN modelling have become more widespread [64, 65], the effect of noise on cellular systems has increasingly been characterised [66–68]. This has not only been important to understand the design principles of native circuits, but also when designing synthetic ones. Indeed, being able to quantify the effect of intrinsic noise is required to optimally design synthetic regulator networks. This will be especially true in the context of sigma factors, as their ability to create orthogonal regulatory systems has gathered interest as synthetic regulators [69–71]. In this article, we have shown that intrinsic cellular noise can generate novel stochastic behaviours in a genetic circuit, and even affect the properties of what is otherwise a purely deterministic behaviour. As the noise properties of more systems are characterised (both experimentally and through models), it will increasingly become clear whether stochastic dynamics, and not deterministic ones, constitute the rule rather than the exception.

## 4 Methods

### 4.1 Model implementation

As a starting point, we use the CRN model implementation of the core σ^B circuit model as introduced in Narula (2016) [45]. Our only modification is that the phosphatase concentration before and after stress have been formalised as model parameters. The model describes the concentration of σ^B, RsbW, and RsbV, as well as the various complexes and dimers they form. It also models the input phosphatase, $P$, as a species of the system.

The strength of the stress is determined by the concentration of the species $P$. In the original model, environmental stress is modelled as a step, where $[P]$ is increased from a low to a higher level at the time of stress (generating a single response pulse). Energy stress is modelled by letting the phosphate concentration adopt a (pre-simulated) gamma-distributed Ornstein-Uhlenbeck process, and then performing reaction rate equation based ODE simulations with this noisy input (creating a stochastic pulsing output). Since we will use the CLE to model system noise, we do not need to add specific noise to the concentration of $P$.

**Table 1. The reactions of the CLE adaptation of Narula model of the σ$^B$ circuit.**

| Description | Reaction | Rate | Propensity |
|---|---|---|---|
| Production | $\varnothing \rightarrow \sigma^B$ | $\nu_0 \frac{1+F[\sigma^B]}{K+[\sigma^B]}$ | $\nu_0 \frac{1+F[\sigma^B]}{K+[\sigma^B]}$ |
| | $\varnothing \rightarrow W$ | $\lambda_W \cdot \nu_0 \frac{1+F[\sigma^B]}{K+[\sigma^B]}$ | $\lambda_W \cdot \nu_0 \frac{1+F[\sigma^B]}{K+[\sigma^B]}$ |
| | $\varnothing \rightarrow V$ | $\lambda_V \cdot \nu_0 \frac{1+F[\sigma^B]}{K+[\sigma^B]}$ | $\lambda_V \cdot \nu_0 \frac{1+F[\sigma^B]}{K+[\sigma^B]}$ |
| Dimerisation | $2W \rightarrow W_2$ | $k_{Bw}$ | $k_{Bw} \frac{[W]^2}{2!}$ |
| Binding | $W_2 + V \rightarrow W_2\text{-}V$ | $k_{B1}$ | $k_{B1}[W_2] \cdot [V]$ |
| | $W_2\text{-}V + V \rightarrow W_2\text{-}V_2$ | $k_{B2}$ | $k_{B2}[W_2\text{-}V] \cdot [V]$ |
| | $W_2 + \sigma^B \rightarrow W_2\text{-}\sigma^B$ | $k_{B3}$ | $k_{B3}[W_2] \cdot [\sigma^B]$ |
| | $P + V^P \rightarrow P\text{-}V^P$ | $k_{B5}$ | $k_{B5}[P] \cdot [V^P]$ |
| Partner switching | $W_2\text{-}\sigma^B + V \rightarrow W_2\text{-}V + \sigma^B$ | $k_{B4}$ | $k_{B4}[W_2\text{-}\sigma^B] \cdot [V]$ |
| | $W_2\text{-}V + \sigma^B \rightarrow W_2\text{-}\sigma^B + V$ | $k_{D4}$ | $k_{D4}[W_2\text{-}V + \sigma^B] \cdot [\sigma^B]$ |
| Dissociation | $W_2 \rightarrow 2W$ | $k_{Dw}$ | $k_{Dw}[W_2]$ |
| | $W_2\text{-}V \rightarrow W_2 + V$ | $k_{D1}$ | $k_{D1}[W_2]$ |
| | $W_2\text{-}RsbV_2 \rightarrow W_2\text{-}V + V$ | $k_{D2}$ | $k_{D2}[W_2]$ |
| | $W_2\text{-}\sigma^B \rightarrow W_2 + \sigma^B$ | $k_{D3}$ | $k_{D3}[W_2]$ |
| | $P\text{-}V^P \rightarrow P + V^P$ | $k_{D5}$ | $k_{D5}[W_2]$ |
| Phosphorylation | $W_2\text{-}V \rightarrow W_2 + V^P$ | $k_{K1}$ | $k_{K1}[W_2\text{-}V]$ |
| | $W_2\text{-}V_2 \rightarrow W_2\text{-}V + V^P$ | $k_{K2}$ | $k_{K2}[W_2\text{-}V_2]$ |
| Dephosphorylation | $P\text{-}V^P \rightarrow P + V$ | $k_P$ | $k_P[P\text{-}V^P]$ |
| Degradation/dilution | $\sigma^B \rightarrow \varnothing$ | $k_{deg}$ | $k_{deg} \cdot [\sigma^B]$ |
| | $W \rightarrow \varnothing$ | $k_{deg}$ | $k_{deg} \cdot [W]$ |
| | $W_2 \rightarrow \varnothing$ | $k_{deg}$ | $k_{deg} \cdot [W_2]$ |
| | $W_2\text{-}\sigma^B \rightarrow \varnothing$ | $k_{deg}$ | $k_{deg} \cdot [W_2\text{-}\sigma^B]$ |
| | $W_2\text{-}V \rightarrow \varnothing$ | $k_{deg}$ | $k_{deg} \cdot [W_2\text{-}V]$ |
| | $W_2\text{-}V_2 \rightarrow \varnothing$ | $k_{deg}$ | $k_{deg} \cdot [W_2\text{-}V_2]$ |
| | $V \rightarrow \varnothing$ | $k_{deg}$ | $k_{deg} \cdot [V]$ |
| | $V^P \rightarrow \varnothing$ | $k_{deg}$ | $k_{deg} \cdot [V^P]$ |
| | $P\text{-}V^P \rightarrow P$ | $k_{deg}$ | $k_{deg} \cdot [P\text{-}V^P]$ |

Here, W stands for RsbW and V for RsbV.

Hence, we can allow all inputs to be step increases in [P]. Formalising this, we let the parameter $p_{init}$ be the concentration of P before stress, and $p_{stress}$ be the concentration of P after stress.

**4.1.1 Reactions.** The model consists of 27 reactions, which are provided in Table 1.

**4.1.2 Parameters.** The model consists of 24 parameters, all of which stay constant throughout a simulation. The parameter values, as they were initially set in [45], are given in Table 2. Throughout the article, the parameters are modified as we investigate how this changes the system's behaviour (exact details of which parameter values are used for each set of simulations can be found in S1–S7 Tables). Finally, we have also introduced an additional 3 parameters to the model: $p_{init}$ is the pre-stress concentration of the phosphatase, $p_{stress}$ is the concentration of the phosphatase in the stressed state, and $\eta$ is the amplitue of noise in the system (using linear scaling of the noise in the CLE).

**4.1.3 SDE and ODE systems.** We will primarily use the CLE to simulate the model. From a set of chemical reactions, it describes a system of SDEs according to

$$dx_i = dt \sum_{j=1}^{m} v_{i,j} \cdot a_j(\bar{x}, t) + \eta \sum_{j=1}^{m} v_{i,j} \sqrt{a_j(\bar{x}, t)} \cdot dW_j \tag{1}$$

where $x_i$ denotes the concentration of the $i$th species, $v_{i,j}$ the net stoichiometric change in the $i$th species as result of the $j$th reaction, and $a_j(\bar{x}, t)$ is the propensity of the $j$th reaction. $dW_j$ denotes a Wiener process with 0 mean and unit variance. Here, the noise scaling parameter is marked as $\eta$.

Due to the additional noise terms, this system becomes too large to yield any useful information when displayed in SDE form. We will instead only write out the ODE system (as generated by the RRE), noting that the SDE system can be unambiguously generated by the CRN's reactions (Table 1) according to the previous equation.

$$
\begin{cases}
\dfrac{d[\sigma^B]}{dt} &= v_0 \dfrac{1 + F[\sigma^B]}{K + [\sigma^B]} + k_{D3}[W_2\text{-}\sigma^B] + k_{B4}[V][W_2\text{-}\sigma^B] - k_{B3}[W_2][\sigma^B] \\[6pt]
&\quad - k_{B4}[W_2\text{-}V][\sigma^B] - k_{deg}[\sigma^B] \\[10pt]
\dfrac{d[W]}{dt} &= \lambda_W \cdot v_0 \dfrac{1 + F[\sigma^B]}{K + [\sigma^B]} + 2k_{Dw}[W_2] - k_{Bw}[W]^2 - k_{deg}[W] \\[10pt]
\dfrac{d[W_2]}{dt} &= k_{D1}[W_2\text{-}V] + k_{D3}[W_2\text{-}\sigma^B] + k_{K1}[W_2\text{-}V] + k_{Bw}\dfrac{[W]^2}{2} - k_{B1}[W_2][V] \\[6pt]
&\quad - k_{B3}[W_2][\sigma^B] - k_{Dw}[W_2] - k_{deg}[W_2] \\[10pt]
\dfrac{d[W_2\text{-}\sigma^B]}{dt} &= k_{B3}[W_2][\sigma^B] + k_{B4}[W_2\text{-}V][\sigma^B] - k_{D4}[W_2\text{-}\sigma^B][V] - k_{D3}[W_2\text{-}\sigma^B] - k_{deg}[W_2\text{-}\sigma^B] \\[10pt]
\dfrac{d[W_2\text{-}V]}{dt} &= k_{D2}[W_2\text{-}V_2] + k_{K2}[W_2\text{-}V_2] + k_{B1}[W_2][V] + k_{B4}[W_2\text{-}V][\sigma^B] - k_{B2}[W_2\text{-}V][V] \\[6pt]
&\quad - k_{D4}[W_2\text{-}\sigma^B][V] - k_{D1}[W_2\text{-}V] - k_{K1}[W_2\text{-}V] - k_{deg}[W_2\text{-}V] \\[10pt]
\dfrac{d[W_2\text{-}V_2]}{dt} &= k_{B2}[W_2\text{-}V][V] - k_{D2}[W_2\text{-}V_2] - k_{K2}[W_2\text{-}V_2] - k_{deg}[W_2\text{-}V_2] \\[10pt]
\dfrac{d[V]}{dt} &= \lambda_V \cdot v_0 \dfrac{1 + F[\sigma^B]}{K + [\sigma^B]} + k_P[P\text{-}V^P] + k_{D1}[W_2\text{-}V] + k_{D2}[W_2\text{-}V_2] + k_{D4}[W_2\text{-}\sigma^B][V] \\[6pt]
&\quad - k_{B1}[W_2][V] - k_{B2}[W_2\text{-}V][V] - k_{B4}[W_2\text{-}V][\sigma^B] - k_{deg}[V] \\[10pt]
\dfrac{d[V^P]}{dt} &= k_{D5}[P\text{-}V^P] + k_{K1}[W_2\text{-}V] + k_{K2}[W_2\text{-}V_2] - k_{B5}[P][V^P] - k_{deg}[V^P] \\[10pt]
\dfrac{d[P]}{dt} &= k_{D5}[P\text{-}V^P] + k_{deg}[P\text{-}V^P] + k_P[P\text{-}V^P] - k_{B5}[P][V^P] \\[10pt]
\dfrac{d[P\text{-}V^P]}{dt} &= k_{B5}[P][V^P] - k_{D5}[P\text{-}V^P] - k_P[P\text{-}V^P] - k_{deg}[P\text{-}V^P]
\end{cases}
$$

**Table 2. The parameters of the CLE adaptation of the Narula model of the σ$^B$ circuit.**

| Parameter | Value | Description |
|---|---|---|
| $v_0$ | 0.4 μM·hr$^{-1}$ | Operon base activity |
| $F$ | 30 | Operon fold change |
| $K$ | 0.2 μM | σ$^B$ binding affinity for operon |
| $\lambda_W$ | 4 | RsbW to σ$^B$ relative production rate |
| $\lambda_V$ | 4.5 | RsbV to σ$^B$ relative production rate |
| $k_{Bw}$ | 3600 μM$^{-1}$hr$^{-1}$ | RsbW dimerisation rate |
| $k_{Dw}$ | 18 hr$^{-1}$ | RsbW dimer dissociation rate |
| $k_{B1}$ | 3600 μM$^{-1}$hr$^{-1}$ | RsbW$_2$ to RsbV binding rate |
| $k_{B2}$ | 3600 μM$^{-1}$hr$^{-1}$ | RsbW$_2$RsbV to RsbV binding rate |
| $k_{B3}$ | 3600 μM$^{-1}$hr$^{-1}$ | RsbW$_2$ to σ$^B$ binding rate |
| $k_{B4}$ | 1800 μM$^{-1}$hr$^{-1}$ | RsbW$_2$σ$^B$ partner switching from σ$^B$ to RsbV rate |
| $k_{B5}$ | 3600 μM$^{-1}$hr$^{-1}$ | Phosphatase to RsbV binding rate |
| $k_{D1}$ | 18 hr$^{-1}$ | RsbW$_2$RsbV dissociation rate |
| $k_{D2}$ | 18 hr$^{-1}$ | RsbW$_2$RsbV$_2$ dissociation rate |
| $k_{D3}$ | 18 hr$^{-1}$ | RsbW$_2$σ$^B$ dissociation rate |
| $k_{D4}$ | 1800 μM$^{-1}$hr$^{-1}$ | RsbW$_2$RsbV partner switching from RsbV to σ$^B$ rate |
| $k_{D5}$ | 18 hr$^{-1}$ | Phosphatase-RsbV dissociation rate |
| $k_{K1}$ | 36 hr$^{-1}$ | Phosphorylation rate (RsbW$_2$RsbV) |
| $k_{K2}$ | 36 hr$^{-1}$ | Phosphorylation rate (RsbW$_2$RsbV$_2$) |
| $k_P$ | 180 hr$^{-1}$ | Dephosphorylation rate |
| $k_{deg}$ | 0.7 hr$^{-1}$ | Degradation/dilution rate |
| $p_{init}$ | 0.001 μM | Phosphatase concentration before stress |
| $p_{stress}$ | variable | Phosphatase concentration during stress |
| $\eta$ | 0.025 | Degree of system noise |

To analyse the impact on the system, the values of a few parameters ($k_{K2}$, $\eta$, $p_{stress}$, $k_P$, $k_{B5}$, and $k_{D5}$) are changed throughout the model investigation. The three last parameters; $p_{stress}$, $p_{init}$, and $\eta$ have been added in this work ($p_{stress}$ and $p_{init}$ were implicitly given in the original model, and formally defined here to aid the model investigation). For some analysis (Fig 3C and onwards), we use the parameter substitutions $p_{prod} = p_{stress} \cdot k_P$ and $p_{frac} = p_{stress}/k_P$.

## 4.2 Modifications to the Narula model

In the CLE adaptation of the Narula model, we can investigate the σ$^B$ response to a step increase in phosphatase input. Here, the upstream pathway activating either RsbTU (environmental stress) or RsbP (energy stress) produces a constant, non-fluctuating, level of phosphatase in response to the stress (while the amount of phosphatase is constant, we note that the relative fraction between phosphatase that is free, or bound to RsbV, fluctuate). In this section, we expand the CLE adaptation of the Narula model, adding fluctuations to the upstream phosphatase input. This allows us to investigate how upstream noise affects the system's response behaviour.

Upstream noise is implemented by assuming that the phosphatase has an active and an inactive state, and that it switches between these according to the law of mass action. This allows us to model the upstream noise through the CLE. By introducing a second CLE noise-scaling parameter for the phosphatase, we can scale the upstream noise separately from that of the core circuit. Through the way we write the new parameters, we can introduce one, $\eta_{amp}$, which scales the amplitude of the noise in the phosphatase, and one, $\eta_{freq}$,

which scales the frequency of the noise in the phosphatase (S18 Fig). Our modified model introduces 1 new species, 3 new reactions, and 2 new parameters.

Similarly to the original Narula model, stress is modelled by increasing the total amount of phosphatase in the system. At the time of stress addition, $[P]$ and $[P_I]$ are both increased by $p_{stress} - p_{init}$ (setting the total amount of phosphatase in the system to $2p_{stress}$, with on average half being in the active state).

**4.2.1 Reactions.** The modified model introduces one new species, $P_I$ (inactive phosphatase), and three reactions. These are the activation and the deactivation of the phosphatase. In addition, phosphatase bound to RsbV can also deactivate (thus dissociating from RsbV) (Table 3).

**4.2.2 Parameters.** The modified model introduces 2 new parameters (Table 4). The new parameter $\eta_{amp}$ scales the noise amplitude in the reactions in Table 3. Meanwhile, the old parameter $\eta$ still scales the noise of all the reactions in Table 1. The new parameter $\eta_{freq}$ scales the rate of phosphatase activation and deactivation. This has no effect on the system's deterministic properties, but will scale the frequency of the fluctuations in the levels of active phosphatase. The parameters $p_{init}$ and $p_{stress}$, while in principle similar to in the previous model, have their properties slightly altered. Instead of representing (before and after stress, respectively) the total amount of phosphatase ($[P_I] + [P] + [P\text{-}V^P]$), they represent the average level of active phosphatase (before and after stress, respectively). The total amount of phosphatase is $2p_{init}$ (before stress) and $2p_{stress}$ (after stress).

**4.2.3 ODE System.** While we will primarily use the SDE system as generated by the CLE, due to the addition of the noise terms, the system becomes too large to yield any useful information when displayed in SDE form. We will instead only write out the ODE system (as generated by the RRE). The SDE system can be unambiguously generated by the CRN's reactions

**Table 3. The reactions added in our modified Narula model.**

| Description | Reaction | Rate | Propensity |
|:---:|:---:|:---:|:---:|
| Activation | $P_I \rightarrow P$ | $\eta_{freq}$ | $\eta_{freq}[P_i]$ |
| Deactivation | $P \rightarrow P_i$ | $\eta_{freq}$ | $\eta_{freq}[P]$ |
|  | $P\text{-}V^P \rightarrow P_i + V^P$ | $\eta_{freq}$ | $\eta_{freq}[P\text{-}V^P]$ |

In addition to these 3 reactions, all the reactions in Table 1 are also a part of the modified Narula model.

**Table 4. The parameters added in our modified Narula model.**

| Parameter | Value | Description |
|:---:|:---:|:---:|
| $\eta_{amp}$ | variable | Upstream noise amplitude |
| $\eta_{freq}$ | variable | Upstream noise frequency |

In addition to these parameters, all the parameters in Table 2 are also a part of the modified Narula model. The system is investigated for a range of $\eta_{amp}$ and $\eta_{freq}$ values, so no particulate values are specified here.

(Tables 1 and 3) through the CLE.

$$\frac{d[\sigma^B]}{dt} = \nu_0 \frac{1+F[\sigma^B]}{K+[\sigma^B]} + k_{D3}[W_2\text{-}\sigma^B] + k_{B4}[V][W_2\text{-}\sigma^B] - k_{B3}[W_2][\sigma^B]$$
$$\qquad\qquad - k_{B4}[W_2\text{-}V][\sigma^B] - k_{deg}[\sigma^B]$$

$$\frac{d[W]}{dt} = \lambda_W \cdot \nu_0 \frac{1+F[\sigma^B]}{K+[\sigma^B]} + 2k_{Dw}[W_2] - k_{Bw}[W]^2 - k_{deg}[W]$$

$$\frac{d[W_2]}{dt} = k_{D1}[W_2\text{-}V] + k_{D3}[W_2\text{-}\sigma^B] + k_{K1}[W_2\text{-}V] + k_{Bw}\frac{[W]^2}{2} - k_{B1}[W_2][V]$$
$$\qquad\qquad - k_{B3}[W_2][\sigma^B] - k_{Dw}[W_2] - k_{deg}[W_2]$$

$$\frac{d[W_2\text{-}\sigma^B]}{dt} = k_{B3}[W_2][\sigma^B] + k_{B4}[W_2\text{-}V][\sigma^B] - k_{D4}[W_2\text{-}\sigma^B][V] - k_{D3}[W_2\text{-}\sigma^B]$$
$$\qquad\qquad - k_{deg}[W_2\text{-}\sigma^B]$$

$$\frac{d[W_2\text{-}V]}{dt} = k_{D2}[W_2\text{-}V_2] + k_{K2}[W_2\text{-}V_2] + k_{B1}[W_2][V] + k_{B4}[W_2\text{-}V][\sigma^B] - k_{B2}[W_2\text{-}V][V]$$
$$\qquad\qquad - k_{D4}[W_2\text{-}\sigma^B][V] - k_{D1}[W_2\text{-}V] - k_{K1}[W_2\text{-}V] - k_{deg}[W_2\text{-}V]$$

$$\frac{d[W_2\text{-}V_2]}{dt} = k_{B2}[W_2\text{-}V][V] - k_{D2}[W_2\text{-}V_2] - k_{K2}[W_2\text{-}V_2] - k_{deg}[W_2\text{-}V_2]$$

$$\frac{d[V]}{dt} = \lambda_V \cdot \nu_0 \frac{1+F[\sigma^B]}{K+[\sigma^B]} + k_P[P\text{-}V^P] + k_{D1}[W_2\text{-}V] + k_{D2}[W_2\text{-}V_2] + k_{D4}[W_2\text{-}\sigma^B][V]$$
$$\qquad\qquad - k_{B1}[W_2][V] - k_{B2}[W_2\text{-}V][V] - k_{B4}[W_2\text{-}V][\sigma^B] - k_{deg}[V]$$

$$\frac{d[V^P]}{dt} = \eta_{freq}[P\text{-}V^P] + k_{D5}[P\text{-}V^P] + k_{K1}[W_2\text{-}V] + k_{K2}[W_2\text{-}V_2] - k_{B5}[P][V^P]$$
$$- k_{deg}[V^P]$$

$$\frac{d[P]}{dt} = \eta_{freq}[P_i] + k_{D5}[P\text{-}V^P] + k_{deg}[P\text{-}V^P] + k_P[P\text{-}V^P] - k_{B5}[P][V^P] - \eta_{freq}[P]$$

$$\frac{d[P_i]}{dt} = \eta_{freq}[P] + \eta_{freq}[P\text{-}V^P] - \eta_{freq}[P_i]$$

$$\frac{d[P\text{-}V^P]}{dt} = k_{B5}[P][V^P] - k_{D5}[P\text{-}V^P] - k_P[P\text{-}V^P] - \eta_{freq}[P\text{-}V^P] - k_{deg}[P\text{-}V^P]$$

## 4.3 Simulations

The models were implemented in the Julia programming language using the Catalyst.jl modelling package [72]. Simulations were carried out using the DifferentialEquations.jl package [73].

For Gillespie simulations, we use the Direct method (Gillespie's direct method), which is the recommended method for small problems [51, 52, 74]. These simulations also require a stepping algorithm (which is used internally to manage the simulation). Since no reactions

were time-dependent, we used the recommended SSAStepper. All Gillespie simulations were performed using the reactions in Table 1, and the parameters described in S3 Table.

For the SDE simulations, we used the implicit Euler-Maruyama method, using fixed time steps [75]. Due to numeric error, differential equations occasionally produce negative species concentration. To prevent these from crashing simulations (due to negative numbers occurring in the CLE's square root noise term), every term within a square root was replaced with its absolute value (e.g. $\sqrt{k_{deg} \cdot [W]}dW_{20}$ would become $\sqrt{|k_{deg} \cdot [W]|}dW_{20}$, where $dW_{20}$ is a Wiener process of the CLE SDE). This has no effect as long as species concentrations stay positive. However, if numeric errors and the noise term push a species into a negative concentration, the absolute value prevents the solver from causing an error due to attempting to take the square root of a negative number [54]. For full details of the simulation, as well as the opportunity to reproduce them, please see the provided scripts (see Section 4.6 for a link to the code).

## 4.4 Bifurcation analysis

Bifurcation analysis was carried out using the BifurcationKit package (which tracks steady states using pseudo-arclength continuation) [76].

## 4.5 Automated behaviour measures

To aid our analysis, we develop four different measures of the system's properties. Of these, the first three are related (measuring the system's ability to generate either behaviour under various circumstances). The last one is a sensitivity measure of whenever either behaviour is sensitive to change in a specified parameter.

**4.5.1 Measure of behavioural magnitude.** Throughout the paper, we wish to determine the magnitude with which either behaviour occurs at a given parameter set $\bar{p} = (k_{Bw}, k_{Dw}, k_{B1}, \ldots)$. To do so, we simulate the system $n$ times, where, $n$ = 50, 100, or 200 (values of $n$ vary from figure to figure and are described in S2, S5 and S7 Tables). Each simulation last 200 hours after the addition of stress (and begins 10 hours before stress). For each simulation we measure:

- The amplitude of the transient pulse. This is measured as the maximum activity of the simulation in the *transient phase* ($t \in [0.0, 5.0]$).

- The maximum asymptotic pulse amplitude. This is measured as the maximum activity of the simulation in the *asymptotic phase* ($t \in [5.0, 200.0]$).

- The mean asymptotic activity. This is measured as the mean activity of the simulation in the *asymptotic phase* ($t \in [5.0, 200.0]$).

Using these, we define the magnitude of the single response pulse behaviour ($M_{srp}(\bar{p})$) and the stochastic pulsing behaviour ($M_{sp}(\bar{p})$) as:

**Single response pulse**: *The amplitude of the transient pulse* divided by *the maximum asymptotic pulse amplitude.*

**Stochastic pulsing**: *The maximum asymptotic pulse amplitude* divided by *the mean asymptotic activity.*

By, for the single response pulse, dividing by *the maximum asymptotic pulse amplitude* we penalise the single response pulse magnitude for the occurrence of stochastic pulses. This ensures that for a single parameter set, the measures cannot both score highly. These measures are illustrated in Fig 2C and 2D.

The five-hour window for the transient phase was chosen to ensure the first (asymptotic) pulse was fully captured. We note that it is possible that this interval might include further pulses, but we still capture this possible sustained pulsing behaviour using our asymptotic phase. To ensure that 5 hours captured the initial pulse, we made an automatic scan of a large number (10, 000) of random parameter sets of our model. None of these exhibited an initial pulse longer than 5 hours.

Finally, to provide additional intuition for these measures, we describe how they will work in two different examples. First, consider a simulation with an oscillatory output. Here, the mean asymptotic activity will typically be half of the maximum asymptotic activity, yielding a stochastic pulsing measure of 2. This value is low compared to a simulation that exhibits stochastic pulsing, where the stochastic pulsing measure often reaches 50. Next, consider a simulation where a single stochastic pulse is exhibited in the asymptotic phase ($t \in [5.0, 200.0]$). This can maximise the magnitude of the stochastic pulsing behaviour, as further pulses will increase the mean asymptotic activity, but only if the further pulses are of lower amplitude, as otherwise the maximum would also increase. We note though that this is only true for the individual simulation. In practice, our measures are always evaluated as the mean over a large ($n > 50$) number of simulations. If one simulation-realisation of a parameter set exhibits a single asymptotic pulse, it is likely that other simulations will exhibit none. For these, the stochastic pulsing measure will equal 1, reducing the overall measure for the parameter set (taken as a mean over all the simulations).

**4.5.2 Measure of the system's ability to robustly generate both behaviours.** Next, we wish to measure to what extent the system can generate both behaviours, as a single parameter is tuned. This is used in S7 Fig to find fixed values for the *core parameters* so that both behaviours can be generated by modulating the *upstream parameters* only. The behaviours should be distinct, that is, the system should clearly exhibit one behaviour to a higher magnitude than the other (and be able to do so for both behaviours). With all other parameters fixed, define $M_{srp}(p)$ and $M_{sp}(p)$ as the magnitude of the two behaviours (both functions of a single parameter $p$). We define

$$M_{srp}^*(p) = \begin{cases} M_{srp}(p) - M_{sp}(p), & M_{srp}(p) > M_{sp}(p) \\ 0, & M_{srp}(p) < M_{sp}(p) \end{cases} \qquad (2)$$

with $M_{sp}^*(p)$ defined similarly. These are the degrees to which either behaviour surpasses the other. We then define our measure (of the degree with which the system can, as a parameter $p$ is varied, distinctly generate both behaviours):

$$D_{srp,sp}(p) = 2 \frac{\sqrt{\left(\int_{p_{start}}^{p_{end}} M_{srp}^*(p)dp\right)\left(\int_{p_{start}}^{p_{end}} M_{sp}^*(p)dp\right)}}{\int_{p_{start}}^{p_{end}} max(M_{srp}^*(p), M_{sp}^*(p))dp} \qquad (3)$$

where $(p_{start}, p_{end})$ is the interval over which we sample the parameter $p$ to calculate the integrals (In practice sums over a dense grid of parameter values are used to estimate the integrals). This measure is illustrated in S6 Fig. Finally, we note that $0 < D_{srp,sp}(p) < 1$.

**4.5.3 Measure of the system's ability to distinctly generate one behaviour.** Next, we wish to measure to what extent the system can generate one behaviour distinctly. That is, to what extent, for a given parameter region, only one behaviour is generated. By optimising this measure, one gets a system with a preference for one of the two behaviours. We do this by modifying the previously defined $D_{srp,sp}(p)$ measure. Again, we will define our measure as a function of only a single parameter, $p$ (in practice it will be carried out using the parameter

$p_{prod}$). Our new measures $D_{srp}(p)$ and $D_{sp}(p)$ can be defined as

$$D_{srp}(p) = \frac{\left(\int_{p_{start}}^{p_{end}} M_{srp}^*(p)dp\right)^2}{\int_{p_{start}}^{p_{end}} max(M_{srp}^*(p), M_{sp}^*(p))dp} \tag{4}$$

with $D_{sp}(p)$ defined similarly. Here, $(p_{start}, p_{end})$ is the interval over which we sample the parameter $p$. Using the terminology in S6 Fig, we have $D_{srp} = \frac{A_{srp}^2}{A_{srp}+A_{sp}+A_{srp.sp}}$. By squaring the numerator, we create a preference for regions in parameter space where the behaviour is prominent (not only prominent in comparison to the other behaviour).

**4.5.4 Measure of the behaviours' sensitivity to the phosphatase parameters.** Finally, we wish to measure to what extent the magnitude of a behaviour changes as we tune a single parameter, $p$. This will be used in Fig 3E and 3F to determine which upstream parameters are important for determining whichever behaviour is generated. Our parameters are sampled at discrete values on a grid, $\{p_i\}_{i=1}^n$. We define our measures $C_{srp}(p, \bar{p})$ and $C_{sp}(p, \bar{p})$ (where $p$ is the values of the model's remaining parameters, which are kept fixed) as:

$$C_{srp}(p, \bar{p}) = (n-1)\sum_{i=1}^{n-1}(M_{srp}(p_{i+1}, \bar{p}) - M_{srp}(p_i, \bar{p}))^2 \tag{5}$$

with $C_{sp}(p, \bar{p})$ defined similarly. The factor $(n-1)$ is added to remove any bias introduced by the density with which the grid is sampled (fewer grid samples mean longer distances between the individual values, these are emphasised by the square, this factor compensates for this). By squaring each value we give preference for sudden dramatic changes, as opposed to gradual change with the parameter.

For simplicity, the $C_{srp}(p, \bar{p})$ and $C_{sp}(p, \bar{p})$ measures shown in Fig 3E and 3F has been normalised by the maximum value of the y-axis.

## 4.6 Code availability

Scripts for generating all of the figures in this article, as well as the simulations on which they are based, can be found at https://gitlab.developers.cam.ac.uk/slcu/teamjl/loman_locke_2023. All scripts are written in the Julia programming language. This enables the definition of a Project.toml file, defining the exact package versions used. This will enable anyone to replicate the exact conditions under which all figures were generated.

## Supporting information

**S1 Fig. Stochastic pulsing is not achievable in the model by tuning $\eta$ alone.** For each combination of stress magnitude ($p_{stress}$) and noise amplitude ($\eta$) four simulations are shown (stress added at red dashed line, $t = 0$). Parameter values and other details on simulation conditions for this figure are described in S1 Table.
(PDF)

**S2 Fig. Only through the tuning of two parameters can oscillations be achieved.** (A) Bifurcation diagrams for the various parameters, each plot shows three diagrams (for the varying stress levels $p_{stress} = 0.05\ \mu M$, $0.20\ \mu M$, and $0.80\ \mu M$). The stars mark the parameter value for the original Narula model. Each x-axis is log10 scaled, and if the parameter's original value is $p_0$, it is varied over the range $(p_0/10, 10p_0)$, corresponding to a tenfold decrease and increase in the target parameter, respectively. Only by tuning $k_{K2}$ or $\lambda_W$ can instability be achieved. For some parameters ($k_{K2}$, $k_P$, and $F$), the curve for $p_{stress} = 0.80\ \mu M$ reaches much larger values

compared to the other two curves, making these hard to distinguish. To avoid figure crowding, periodic orbits are not displayed in these diagrams, however, they are instead shown in S3 Fig. (B) Bifurcation diagram for the parameter $p_{stress}$ (the magnitude of the stress) over the interval (0.1 μM,10.0 μM), with the x-axis log10 scaled. Instability cannot be produced by tuning $p_{stress}$ only. Parameter values and other details on simulation conditions for this figure are described in S1 Table.
(PDF)

**S3 Fig. Bifurcation diagrams with periodic orbits shown.** Four of the bifurcation diagrams in S2 Fig display periodic orbits, these are shown in more details here. The stars mark the parameters' values for the original Narula model and all x-axes are log10 scaled. (A) Bifurcation diagram with respect to the parameter $k_{K2}$. (B-D) Bifurcation diagram with respect to the parameter $\lambda_V$, for three different values of $p_{stress}$. Note that for these the y-axes are log10 scaled (to help show the periodic orbits more clearly), unlikely in S2 Fig where they are linearly scaled. Parameter values and other details on simulation conditions for this figure are described in S1 Table.
(PDF)

**S4 Fig. Both behaviours can be reproduced even as we remove all noise from upstream reactions.** Here we test whether removing all noise from phosphatase reactions can still allow both response dynamics. In Fig 2 we demonstrate that both system behaviours can be recreated even as phosphatase levels remain constant (unlike in [45] which assumed noisy concentrations of $P$). To do this, we use the CLE. The CLE adds noise to reaction channels, rather than species concentrations (with the former affecting the latter). This means that there's still noise in $P$ due to noise in the reactions involving $P$. (A,B) Here we remove noise from all reactions involving $P$ (and components containing $P$). To do this, we use our modified Narula model that allows us to tune upstream noise (Section 4.2) and set $\eta_{amp}$ = 0.0. Next, we recreate both the single response pulse (A, $p_{prod}$ = 50.0, $\eta$ = 0.01) and stochastic pulsing (B, $p_{prod}$ = 25.0, $\eta$ = 0.09) behaviours. This demonstrates that both behaviours can be generated by the core circuit, and are not dependent on any form of upstream noise. Stress is added at red dashed lines ($t$ = 0) and each plot shows four simulations. Full parameter sets for this figure are described in S6 Table.
(PDF)

**S5 Fig. The stochastic pulses in σ$^B$ are preceded by a reduction in total RsbW concentration.** (A) In the time leading up to a stochastic pulse, the total amount of σ$^B$ (blue, scale marked at the left) and RsbW (blue, scale marked at the right) is plotted. Four simulations are shown. (B) For the same four simulations, the system is shown in σ$^B$-RsbW phase space. The state at the beginning of the simulation is marked with a blue dot. Just before the pulse is initiated (σ$^B$ levels start to rise), the total RsbW concentration dips. Although harder to see, the dip can also be distinguished in (A). Parameter values and other details on simulation conditions for this figure are described in S1 Table.
(PDF)

**S6 Fig. A measure of the system's ability to generate both behaviours distinctly as a single parameter is tuned.** We find the two functions $M_{srp}(p)$ and $M_{sp}(p)$ of our target parameter (this example used $p_{stress}$) (Section 4.5.2). We define three areas: $A_{srp}$ is the area which is beneath the $M_{srp}(p)$ curve but above $M_{sp}(p)$, with $A_{sp}$ defined similarly. We also define $A_{srp,sp}$ as the area which is beneath both curves. Finally, our measure is defined as

$D_{srp,sp}(p) = \frac{\sqrt{A_{srp} \cdot A_{sp}}}{A_{srp} + A_{srp,sp} + A_{sp}}$. Parameter values and other details on simulation conditions for this

figure are described in S2 Table.
(PDF)

**S7 Fig. Heatmap showing which combinations of $k_{K2}$ and $\eta$ enables the system to optimally generate both behaviours.** The function $D_{srp,sp}(p_{stress}, \bar{p})$ measures the system's ability to distinctly generate both behaviours, while varying the parameter $p_{stress}$ only (Section 4.5.2). The optimal value, $(k_{K2}, \eta) = (7.0hr^{-1}, 0.025)$, is found close to the bottom left corner (light green dot). We chose the upper limit of $\eta$ (0.15) as beyond this point behaviours start to become obscured by noise levels. Parameter values and other details on simulation conditions for this figure are described in S2 Table.
(PDF)

**S8 Fig. An optimised parameter set is able to generate both behaviours by varying $p_{stress}$ only.** (A,B) The system's response to stress (red dashed line, $t = 0$), for $(k_{K2}, \eta) = (7.0hr^{-1}, 0.025)$ and $p_{stress} = 0.24$ μM (A) and $p_{stress} = 0.28$ μM (B). Each plot shows four simulations. Parameter values and other details on simulation conditions for this figure are described in S1 Table.
(PDF)

**S9 Fig. The shape of the two behaviours' regions of occurrence is stable as $k_{B5}$ and $k_{D5}$ are changed.** Heatmaps showing the two behaviours occurrences in $k_{B5}$-$k_{D5}$-space. In all plots, the behaviours region of occurrence is similar to a $k_P \cdot p_{stress} = C$ curve. Parameter values and other details on simulation conditions for this figure are described in S2 Table.
(PDF)

**S10 Fig. Heatmaps describing the magnitude of the single response pulse behaviour for various values of $k_{B5}$ and $k_{D5}$.** Each heatmap describes the behaviour's magnitude as the parameters $p_{prod}$ (x-axis) and $p_{frac}$ (y-axis) are varied. A total of 36 heatmaps are plotted and placed in a 6x6 grid for a range of values of $k_{B5}$ and $k_{D5}$. There is a distinct spike in magnitude as $p_{prod}$ is varied. Changes to $p_{frac}$, $p_{prod}$, and $k_{D5}$ all have some effect on the magnitude, but not as distinct as changes to $p_{prod}$. Parameter values and other details on simulation conditions for this figure are described in S5 Table.
(PDF)

**S11 Fig. Heatmaps describing the magnitude of the stochastic pulsing response behaviour for various values of $k_{B5}$ and $k_{D5}$.** Each heatmap describes the behaviour's magnitude as the parameters $p_{prod}$ (x-axis) and $p_{frac}$ (y-axis) are varied. A total of 36 heatmaps are plotted and placed in a 6x6 grid for a range of values of $k_{B5}$ and $k_{D5}$. There is a distinct spike in magnitude as $p_{prod}$ is varied. Changes to $p_{frac}$, $p_{prod}$, and $k_{D5}$ all have some effect on the magnitude, but not as distinct as changes to $p_{prod}$. Parameter values and other details on simulation conditions for this figure are described in S5 Table.
(PDF)

**S12 Fig. Heatmaps describing the magnitude of the single response pulse behaviour for various values of $p_{prod}$ and $p_{frac}$.** Each heatmap describes the behaviour's magnitude as the parameters $k_{D5}$ (x-axis) and $k_{B5}$ (y-axis) are varied. A total of 36 heatmaps are plotted and placed in a 6x6 grid for a range of values of $p_{prod}$ and $p_{frac}$. Only for one value of $p_{prod}$ do changes in the other parameters have a major effect on the behaviour's magnitude. Parameter values and other details on simulation conditions for this figure are described in S5 Table.
(PDF)

**S13 Fig. Heatmaps describing the magnitude of the stochastic pulsing response behaviour for various values of $p_{prod}$ and $p_{frac}$.** Each heatmap describes the behaviour's magnitude as the parameters $k_{D5}$ (x-axis) and $k_{B5}$ (y-axis) are varied. A total of 36 heatmaps are plotted and placed in a 6x6 grid for a range of values of $p_{prod}$ and $p_{frac}$. Only for one value of $p_{prod}$ do changes in the other parameters have a major effect on the behaviour's magnitude. Parameter values and other details on simulation conditions for this figure are described in S5 Table. (PDF)

**S14 Fig. Heatmaps describing the magnitude of the single response pulse behaviour for various values of $p_{frac}$ and $k_{D5}$.** Each heatmap describes the behaviour's magnitude as the parameters $p_{prod}$ (x-axis) and $k_{B5}$ (y-axis) are varied. A total of 36 heatmaps are plotted and placed in a 6x6 grid for a range of values of $p_{frac}$ and $k_{D5}$. There is a distinct spike in magnitude as $p_{prod}$ is varied. Changes to $p_{frac}$, $p_{prod}$, and $k_{D5}$ have little effect on the behaviour's magnitude. Parameter values and other details on simulation conditions for this figure are described in S5 Table. (PDF)

**S15 Fig. Heatmaps describing the magnitude of the stochastic pulsing response behaviour for various values of $p_{frac}$ and $k_{D5}$.** Each heatmap describes the behaviour's magnitude as the parameters $p_{prod}$ (x-axis) and $k_{B5}$ (y-axis) are varied. A total of 36 heatmaps are plotted and placed in a 6x6 grid for a range of values of $p_{frac}$ and $k_{D5}$. There is a distinct spike in magnitude as $p_{prod}$ is varied. Changes to $p_{frac}$, $p_{prod}$, and $k_{D5}$ have little effect on the behaviour's magnitude. Parameter values and other details on simulation conditions for this figure are described in S5 Table. (PDF)

**S16 Fig. Fig 4 with a system bifurcation diagram added.** Subplot A is identical to subplot A of Fig 4. Subplots C-N are identical to subplots B-M of Fig 4. Please see Fig 4 for legends of these subplots. (B) Bifurcation diagram, showing the system's steady state as a function of $p_{prod}$. After a region of inactivity, the steady state becomes unstable (implying a limit cycle). As $p_{prod}$ increases further, the steady state becomes stable, with an increasing concentration that eventually saturates. The transition in the bifurcation diagram corresponds to the transition in C-N (stability at an inactive state, a limit cycle, stability at an active state). We note that the bifurcation diagram is computed from the deterministic (ODE) system, while subfigures A, and C-N all use the stochastic (SDE) system. Adding noise to a non-linear system affects the parameter values at which stability occurs. This explains that the region of instability in (B) occurs for different $p_{prod}$ values as compared to A. While the deterministic bifurcation analysis in B cannot be directly translated to our stochastic system, it is still worth noting this similarity between the two cases. Parameter values and other details on simulation conditions for this figure are described in S5 Table. (PDF)

**S17 Fig. The behavioural transition, as $p_{prod}$ is varied, in Fig 4 can be recreated using the Gillespie algorithm.** In Fig 4 we observed that the system undergoes a behavioural transition as $p_{prod}$ is increased from small to high. For small values of $p_{prod}$ the system is inactive. As $p_{prod}$ is increased the system exhibits, in order, single response pulse, stochastic pulsing, oscillating, and persistent activity, behaviours. Here, we recreate the same transition using Gillespie simulations. While we in Fig 4 vary the parameter $p_{prod}$, we never introduce this parameter substitution for the Gillespie approach. We instead vary the parameter $p_{stress}$. However, since $p_{prod} = p_{stress} \cdot k_P$, the two transitions should be equivalent. (A-L) Gillespie simulation of the Narula model for various values of $p_{stress}$. Each frame contains 4 simulations, and stress is added at the

red dashed lines ($t = 0$). The behavioural transition from Fig 4 is recreated. Parameter values and other details on simulation conditions for this figure are described in S3 Table.
(PDF)

**S18 Fig. In the modified Narula model, two parameters ($\eta_{amp}$ and $\eta_{freq}$), allow us to scale the amplitude and frequency of the noise in the input phosphatase.** By splitting the phosphatase into an active state ($P$) and an inactive state ($P_i$), with only the active state being able to form the P-VP complex, we can introduce fluctuations in phosphatase levels. Before the addition of stress, the total amount of phosphatase ($[P_i] + [P] + [P\text{-}VP]$) is $2p_{init}$ (with, on average, at every time point, half being in the active state). At the time of stress onset (here at the red dashed line at $t = 0$), the amount of phosphatase is increased to $2p_{stress}$ (again, with on average half being active). The frequency of the fluctuations scales with the parameter $\eta_{freq}$. The amplitude of the fluctuations scales with the parameter $\eta_{amp}$. Parameter values and other details on simulation conditions for this figure are described in S7 Table.
(PDF)

**S1 Table. Parameter values for simulation of the CLE adaptation of the Narula model.** For each figure where the model is simulated, the parameter values used for the simulations are marked. If not marked, the following parameter values are used: $k_{Bw} = 3600$ μM$^{-1}$hr$^{-1}$, $k_{Dw} = 18$ hr$^{-1}$, $k_{B1} = 3600$ μM$^{-1}$hr$^{-1}$, $k_{B2} = 3600$ μM$^{-1}$hr$^{-1}$, $k_{B3} = 3600$ μM$^{-1}$hr$^{-1}$, $k_{B4} = 1800$ μM$^{-1}$hr$^{-1}$, $k_{B5} = 3600$ μM$^{-1}$hr$^{-1}$, $k_{D1} = 18$ hr$^{-1}$, $k_{D2} = 18$ hr$^{-1}$, $k_{D3} = 18$ hr$^{-1}$, $k_{D4} = 1800$ μM$^{-1}$hr$^{-1}$, $k_{D5} = 18$ hr$^{-1}$, $k_{K1} = 36$ hr$^{-1}$, $k_{K2} = 36$ hr$^{-1}$, $k_P = 180$ hr$^{-1}$ hr$^{-1}$, $k_{Deg} = 0.7$ hr$^{-1}$, $v_0 = 0.4$ μM$^{-1}$hr$^{-1}$, $F = 30$, $K = 0.2$ μM, $\lambda_W = 4$, $\lambda_V = 4.5$, $\eta = 0.05$, $p_{init} = 0.001$ μM, $p_{stress} = 0.4$ μM. Finally, in certain figures, some parameter values are varied as marked on the figures. Which parameters are varied across each is marked in the last column.
(PDF)

**S2 Table. Parameter values for parameter scans of the CLE adaptation of the Narula model.** For each figure where we perform parameter scans of the model's behaviour, the parameter values used are marked. If not marked, the following parameter values are used: $k_{Bw} = 3600$ μM$^{-1}$hr$^{-1}$, $k_{Dw} = 18$ hr$^{-1}$, $k_{B1} = 3600$ μM$^{-1}$hr$^{-1}$, $k_{B2} = 3600$ μM$^{-1}$hr$^{-1}$, $k_{B3} = 3600$ μM$^{-1}$hr$^{-1}$, $k_{B4} = 1800$ μM$^{-1}$hr$^{-1}$, $k_{B5} = 3600$ μM$^{-1}$hr$^{-1}$, $k_{D1} = 18$ hr$^{-1}$, $k_{D2} = 18$ hr$^{-1}$, $k_{D3} = 18$ hr$^{-1}$, $k_{D4} = 1800$ μM$^{-1}$hr$^{-1}$, $k_{D5} = 18$ hr$^{-1}$, $k_{K1} = 36$ hr$^{-1}$, $k_{K2} = 36$ hr$^{-1}$, $k_P = 180$ hr$^{-1}$, $k_{Deg} = 0.7$ hr$^{-1}$, $v_0 = 0.4$ μM$^{-1}$hr$^{-1}$, $F = 30$, $K = 0.2$ μM, $\lambda_W = 4$, $\lambda_V = 4.5$, $\eta = 0.05$, and $p_{init} = 0.001$ μM. The third column denotes how many simulations ($n$) are performed for each parameter combination. Finally, in certain figures, some parameter values are varied as marked on the figures. Which parameters are varied across each is marked in the last column.
(PDF)

**S3 Table. Parameter values for Gillespie simulation of the Narula model.** Here, Ms stand for Molecules. For each figure where the Narula model is simulated using the Gillespie algorithm, the parameter values used for the simulations are marked. If not marked, the following parameter values are used: $k_{Bw} = 3600$ Molecules$^{-1}$hr$^{-1}$, $k_{Dw} = 18$ hr$^{-1}$, $k_{B1} = 3600$ Molecules$^{-1}$hr$^{-1}$, $k_{B2} = 3600$ Molecules$^{-1}$hr$^{-1}$, $k_{B3} = 3600$ Molecules$^{-1}$hr$^{-1}$, $k_{B4} = 1800$ Molecules$^{-1}$hr$^{-1}$, $k_{B5} = 3600$ Molecules$^{-1}$hr$^{-1}$, $k_{D1} = 18$ Molecules$^{-1}$, $k_{D2} = 18$ hr$^{-1}$, $k_{D3} = 18$ hr$^{-1}$, $k_{D4} = 1800$ Molecules$^{-1}$hr$^{-1}$, $k_{D5} = 18$ hr$^{-1}$, $k_{K1} = 36$ hr$^{-1}$, $k_P = 180$ hr$^{-1}$, $\lambda_W = 4$, $\lambda_V = 4.5$, and $p_{init} = 0$ Molecules. Finally, in S18 Fig, the value of $p_{stress}$ is varied as marked in the figure. This is designated by, in this table, putting $p_{stress}$ in the "Varied parameters" column.
(PDF)

**S4 Table. Parameter values for simulation of the CLE adaptation of the Narula model with parameter substitution.** For each figure where the model is simulated, the parameter values used for the simulations are marked. If not marked, the following parameter values are used: $k_{Bw}$ = 3600 μM$^{-1}$hr$^{-1}$, $k_{Dw}$ = 18 hr$^{-1}$, $k_{B1}$ = 3600 μM$^{-1}$hr$^{-1}$, $k_{B2}$ = 3600 μM$^{-1}$hr$^{-1}$, $k_{B3}$ = 3600 μM$^{-1}$hr$^{-1}$, $k_{B4}$ = 1800 μM$^{-1}$hr$^{-1}$, $k_{B5}$ = 3600 μM$^{-1}$hr$^{-1}$, $k_{D1}$ = 18 hr$^{-1}$, $k_{D2}$ = 18 hr$^{-1}$, $k_{D3}$ = 18 hr$^{-1}$, $k_{D4}$ = 1800 μM$^{-1}$hr$^{-1}$, $k_{D5}$ = 18 hr$^{-1}$, $k_{K1}$ = 36 hr$^{-1}$, $k_{K2}$ = 36 hr$^{-1}$, $k_{Deg}$ = 0.7 hr$^{-1}$, $v_0$ = 0.4 μM$^{-1}$hr$^{-1}$, $F$ = 30, $K$ = 0.2 μM, $\lambda_W$ = 4, $\lambda_V$ = 4.5, $\eta$ = 0.05, $p_{init}$ = 0.001 μM, and $p_{frac}$ = 100 μM hr$^1$. Finally, in certain figures, some parameter values are varied as marked on the figures. Which parameters are varied across each is marked in the last column.
(PDF)

**S5 Table. Parameter values for parameter scans of the CLE adaptation of the Narula model with parameter substitution.** For each figure where we perform parameter scans of the model's behaviour, the parameter values used are marked. If not marked, the following parameter values are used: $k_{Bw}$ = 3600 μM$^{-1}$hr$^{-1}$, $k_{Dw}$ = 18 hr$^{-1}$, $k_{B1}$ = 3600 μM$^{-1}$hr$^{-1}$, $k_{B2}$ = 3600 μM$^{-1}$hr$^{-1}$, $k_{B3}$ = 3600 μM$^{-1}$hr$^{-1}$, $k_{B4}$ = 1800 μM$^{-1}$hr$^{-1}$, $k_{B5}$ = 3600 μM$^{-1}$hr$^{-1}$, $k_{D1}$ = 18 hr$^{-1}$, $k_{D2}$ = 18 hr$^{-1}$, $k_{D3}$ = 18 hr$^{-1}$, $k_{D4}$ = 1800 μM$^{-1}$hr$^{-1}$, $k_{D5}$ = 18 hr$^{-1}$, $k_{K1}$ = 36 hr$^{-1}$, $k_{K2}$ = 36 hr$^{-1}$, $k_{Deg}$ = 0.7 hr$^{-1}$, $v_0$ = 0.4 μM$^{-1}$hr$^{-1}$, $F$ = 30, $K$ = 0.2 μM, $\lambda_W$ = 4, $\lambda_V$ = 4.5, $\eta$ = 0.05, $p_{init}$ = 0.001 μM, and $p_{frac}$ = 100 μM hr$^1$. The third column denotes how many simulations ($n$) are performed for each parameter combination. These simulations are then used to determine behavioural magnitudes (larger $n$ yields smoother plots). Finally, in certain figures, some parameter values are varied as marked on the figures. Which parameters are varied across each is marked in the last column.
(PDF)

**S6 Table. Parameter values for simulation of the modified Narula model.** For each figure where the modified Narula model is simulated, the parameter values used for the simulations are marked. If not marked, the following parameter values are used: $k_{Bw}$ = 3600 μM$^{-1}$hr$^{-1}$, $k_{Dw}$ = 18 hr$^{-1}$, $k_{B1}$ = 3600 μM$^{-1}$hr$^{-1}$, $k_{B2}$ = 3600 μM$^{-1}$hr$^{-1}$, $k_{B3}$ = 3600 μM$^{-1}$hr$^{-1}$, $k_{B4}$ = 1800 μM$^{-1}$hr$^{-1}$, $k_{B5}$ = 3600 μM$^{-1}$hr$^{-1}$, $k_{D1}$ = 18 hr$^{-1}$, $k_{D2}$ = 18 hr$^{-1}$, $k_{D3}$ = 18 hr$^{-1}$, $k_{D4}$ = 1800 μM$^{-1}$hr$^{-1}$, $k_{D5}$ = 18 hr$^{-1}$, $k_{K1}$ = 36 hr$^{-1}$, $k_{K2}$ = 36 hr$^{-1}$, $k_{Deg}$ = 0.7 hr$^{-1}$, $v_0$ = 0.4 μM$^{-1}$hr$^{-1}$, $F$ = 30, $K$ = 0.2 μM, $\lambda_W$ = 4, $\lambda_V$ = 4.5, $\eta$ = 0.05, $p_{init}$ = 0.001 μM, $p_{frac}$ = 100 μM hr$^1$, $\eta_{amp}$ = 0.05, and $\eta_{freq}$ = 1. Finally, in certain figures, some parameter values are varied as marked on the figures. Which parameters are varied across each is marked in the last column.
(PDF)

**S7 Table. Parameter values for parameter scans of the modified Narula model.** For each figure where we perform parameter scans of the modified Narula model's behaviour, the parameter values used are marked. If not marked, the following parameter values are used: $k_{Bw}$ = 3600 μM$^{-1}$hr$^{-1}$, $k_{Dw}$ = 18 hr$^{-1}$, $k_{B1}$ = 3600 μM$^{-1}$hr$^{-1}$, $k_{B2}$ = 3600 μM$^{-1}$hr$^{-1}$, $k_{B3}$ = 3600 μM$^{-1}$hr$^{-1}$, $k_{B4}$ = 1800 μM$^{-1}$hr$^{-1}$, $k_{B5}$ = 3600 μM$^{-1}$hr$^{-1}$, $k_{D1}$ = 18 hr$^{-1}$, $k_{D2}$ = 18 hr$^{-1}$, $k_{D3}$ = 18 hr$^{-1}$, $k_{D4}$ = 1800 μM$^{-1}$hr$^{-1}$, $k_{D5}$ = 18 hr$^{-1}$, $k_{K1}$ = 36 hr$^{-1}$, $k_{K2}$ = 36 hr$^{-1}$, $k_{Deg}$ = 0.7 hr$^{-1}$, $v_0$ = 0.4 μM$^{-1}$hr$^{-1}$, $F$ = 30, $K$ = 0.2 μM, $\lambda_W$ = 4, $\lambda_V$ = 4.5, $\eta$ = 0.05, $p_{init}$ = 0.001 μM, $p_{frac}$ = 100 μM hr$^1$, $\eta_{amp}$ = 0.05, and $\eta_{freq}$ = 1. The third column denotes how many simulations ($n$) are performed for each parameter combination. These simulations are then used to determine behavioural magnitudes (larger $n$ yields smoother plots). Finally, in certain figures, some parameter values are varied as marked on the figures. Which parameters are varied across each is marked in the last column.
(PDF)

## Acknowledgments

We are grateful for the high performance computing resources provided by the Cambridge Service for Data Driven Discovery. We thank Dr. Katie Abley and Dr Christian Schwall (University of Cambridge) for critical reading of the manuscript.

## Author Contributions

**Conceptualization:** Torkel E. Loman, James C. W. Locke.

**Data curation:** Torkel E. Loman.

**Formal analysis:** Torkel E. Loman.

**Funding acquisition:** James C. W. Locke.

**Investigation:** Torkel E. Loman.

**Methodology:** Torkel E. Loman.

**Project administration:** James C. W. Locke.

**Software:** Torkel E. Loman.

**Supervision:** James C. W. Locke.

**Validation:** Torkel E. Loman.

**Visualization:** Torkel E. Loman.

**Writing – original draft:** Torkel E. Loman, James C. W. Locke.

**Writing – review & editing:** Torkel E. Loman, James C. W. Locke.

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
