## [Decision Letter · Decision Letter 0]

11 Nov 2022

Dear Dr. Locke,

Thank you very much for submitting your manuscript "The σ^B^ alternative sigma factor circuit modulates noise to generate different types of pulsing dynamics" for consideration at PLOS Computational Biology.

As with all papers reviewed by the journal, your manuscript was reviewed by members of the editorial board and by several independent reviewers. While all reviewers acknowledge the potential relevance of this work (to varying degrees), there is considerable confusion among the reviewers regarding technical and other aspects. While some of reviewer #2’s criticism may be somewhat exaggerated, their more fundamental concerns partially overlap with those of reviewer #1 and seem valid. Nevertheless, if you feel that you can convincingly address these concerns, we are willing to consider a heavily revised version of this manuscript. If you decide to submit a revised manuscript, it would be important to include a response to all reviewer concerns.

We cannot make any decision about publication until we have seen the revised manuscript and your response to the reviewers' comments. Your revised manuscript is also likely to be sent to reviewers for further evaluation.

Sincerely,

Tobias Bollenbach

Academic Editor

PLOS Computational Biology

Mark Alber

Section Editor

PLOS Computational Biology

While all reviewers acknowledge the potential relevance of this work (to varying degrees), there is considerable confusion among the reviewers regarding technical and other aspects. While some of reviewer #2’s criticism may be somewhat exaggerated, their more fundamental concerns partially overlap with those of reviewer #1 and seem valid. Nevertheless, if you feel that you can convincingly address these concerns, we are willing to consider a heavily revised version of this manuscript. If you decide to submit a revised manuscript, it would be important to include a response to all reviewer concerns.

Reviewer's Responses to Questions

**Comments to the Authors:**

Reviewer #1: The paper by Loman and Locke presents a simulation-based study to understand how different types of pulsing dynamics may be modulated by noise in the σB alternative sigma factor circuit. They show that whether the response is stochastic pulsing or single response pulse is determined by the degree with which the input pathway activates the core circuit and also by the stochastic properties of the input pathway. The results are certainly interesting. However I have some concerns about the type of noise captured by their study (they ignore the major sources of extrinsic noise), and some details about the simulation methods appear missing.

1. In the Introduction they write "Traditionally such models have been deterministic in nature, however, due to low molecular copy numbers, cellular networks are often stochastic". I think this picture is now somewhat outdated. The noise they describe is intrinsic noise but many studies have shown that extrinsic sources of noise are at least equally important, if not more important than intrinsic noise. So for e.g. variation amongst cells due to cell size, doubling time, etc is a leading cause of why cellular dynamics is perceived to be stochastic. Indeed the literature in the past decade has moved towards incorporating explicitly various sources of extrinsic noise. A few papers investigating such sources:

Shahrezaei, Vahid, Julien F. Ollivier, and Peter S. Swain. "Colored extrinsic fluctuations and stochastic gene expression." Molecular systems biology 4.1 (2008): 196.

Benzinger, Dirk, and Mustafa Khammash. "Pulsatile inputs achieve tunable attenuation of gene expression variability and graded multi-gene regulation." Nature communications 9.1 (2018): 1-10.

Perez-Carrasco et al. "Effects of cell cycle variability on lineage and population measurements of messenger RNA abundance." Journal of the Royal Society Interface 17.168 (2020): 20200360.

Ietswaart, Robert, et al. "Cell-size-dependent transcription of FLC and its antisense long non-coding RNA COOLAIR explain cell-to-cell expression variation." Cell Systems 4.6 (2017): 622-635.

The present study is based on the use of the chemical Langevin equation which has a system size parameter in it -- they consider this to be a parameter that scales the amplitude of the noise in the system but actually the system size is the cell volume (see van Kampen's book) and hence this varies from cell to cell. The experimental confirmation of the predictions they are making will necessarily involve a population of cells with an associated cell size distribution and even if one can follow a single cell, its size would vary with time. Its hence unclear to me how how much the addition of these relevant types of extrinsic noise will impact the present results. I suggest they modify their SDE approach to account for some of the relevant sources e.g. cell size variation in a population.

2. It should be explicitly mentioned that the CLE is a valid approach in the limit of small noise (I could not find this mentioned). It is known that if the CLE simulations approach zero molecule numbers then there is a possibility that the numbers go negative which is unrealistic. Often an artificial reflective boundary is thus implemented such that if the molecule number goes negative then it is set back to zero. This can lead to artifactual effects however in the simulation output. Have the authors of the present study employed such a reflective boundary?

3. According to Table 1, dilution is modelled via effective first-order reactions. While this is commonly done, it has in recent years been shown to be inaccurate when compared to a model which explicitly models dilution, i.e. models cell division and the partitioning of molecules between daughter cells. See for e.g. Beentjes, et al. "Exact solution of stochastic gene expression models with bursting, cell cycle and replication dynamics." Physical Review E 101.3 (2020): 032403. I am not suggesting that the authors modify their model to take into account this more realistically but at least this should be adequately discussed as a limitation of the present model emphasizing that effective dilution reactions underestimate the true size of the noise from partitioning at cell division; as well, some discussion should be added on how this might affect the present results.

4. On P. 13 in the equation for [W_2] there is a missing bracket in the fourth term on the RHS. On P. 8 "simualte" should be "simulate".

Reviewer #2: In this manuscript the authors analyse a model of bacterial stress response in the presence of noise. In particular, the authors show that a variant of a previously published model of the alternative sigma factor SigB response can generate two different responses (single pulse response vs stochastic pulsing) depending on the input stress.

Their main finding is that their stochastic model can give rise to either type of response for deterministic stress input. This extends the results reported previously in the literature which analysed a deterministic model which could only generate single pulse responses when subjected to deterministic step-change in input stress.

Bacterial stress response and the effect of noise on biochemical reaction pathways is an important area of much recent experimental and theoretical interest. However, I have several misgivings about the presented research.

1) It is unclear what the wider scientific contribution to the field is. Before the authors’ current work, the field had analysed the deterministic version of their model which generated stochastic responses only when the input was stochastic. The fact that a deterministic model can only give deterministic outputs to deterministic inputs is of course a given. In the current manuscript the authors added some noise to the system and showed that now even a deterministic input can give rise to stochastic output behaviour. While it is a rewarding exercise to show this explicitly, I am sceptical this result significantly contributes to the field because the result comes as no surprise and is expected behaviour. In addition, even on a technical level it is unclear to me that a significant technical contribution was made. That is because of the way the authors add noise to the system as detailed next.

2) The authors add noise by analysing Langevin type dynamics with a global ad hoc noise strength parameter that affects all components. This seems like an unrealistic/unreliable approach to elucidate the actual effect of stochastic noise in the biochemical response dynamics. The authors comment on this very briefly in their introduction and support their choice by an argument of convenience. I find the lack of scientific justification unconvincing. Because the author’s approach unavoidably randomizes the immediate effect of the input stress in the signal, I cannot see how the authors exclude the possibility that their observed dynamics is essentially equivalent to the previously observed one: a deterministic input that is immediately randomized in the first signalling event is essentially equivalent to adding noise to the input in the first place (which had been analysed previously). All the noise added to the remaining steps might not have played any role.

3) The work does not take into account that multiple sigma factors compete for a limited pool of RNA polymerases. Such competition effects have been shown to play an important role in bacterial stress response. In fact, one of the main points of the previous analysis of the deterministic model was to determine the effect of competition. Since the authors base their work on this previous model, they ought to include the same type of analysis.

4) The manuscript is often unclear on crucial technical details to the point that the overall presentation is too imprecise for a scientific publication. I will detail some of these issues explicitly below as well as some of the minor typographic/presentation errors I noticed throughout.

L246 The authors write “We use a (slightly altered) implementation of the core SigB circuit model implemented in [41]" Exactly how is this model altered? There is a sentence that follows explaining the model, but it only talks about the difference that P is a species. Is that all? Please clarify explicitly and explain in words how the model is different.

If P is now a species in their model, I did not understand its dynamics. The authors write "The strength of the stress is determined by the concentration of the species P." How is P increased? Should be specified/explained. This was confusing because reading through the results it seemed like P was *not* a species but just an input parameter that was instantaneously changes.

As is, the presentation was very confusing, for example, the authors write "We will instead perform CLE based SDE simulations of the model. All inputs will be step increases in [P]." Which seems to contradict the model definition which suggests that noise was added to all species. Was P not a model species despite the statement that their model “models the input phosphatase, P, as a species of the system” or was it a species that was not subject to noise? While I am sure the authors are entirely clear in the way they set up their model the way it is described in this manuscript is not precise.

L266 The authors introduce a global noise parameter that affects the dynamics of all species. Later they vary this noise parameter, for example in Fig 1 eta ranges from 0 to: 0 0.15. It is unclear what this magnitude corresponds to in biochemical terms. Is 0.1 small or large?

L277 The authors write "In this section, we expand the Narula model, adding fluctuations to the upstream phosphatase input." I am confused by this statement as the Narula et al. work specifically considered noisy fluctuations in phosphatase levels.

L84 "To implement a stochastic CRN, we used the chemical Langevin equation (CLE) [42] We noted [sic] that it is also possible to simulate the CRN using the Gillespie algorithm [43, 44]. However, this interpretation does not allow for tuning of the system’s degree of noise independently of its other properties."

This is an argument based on convenience instead of scientific merit. This modelling choice is potentially problematic and should be discussed in detail. In fact, the authors included a supplementary figure (S1) that seems to show actual Gillespie simulations of a stochastic version of their system. I could not find any explanation of this figure/dynamics in the main results. Remarkably, the stochastic realizations of the Gillespie simulations did not seem exhibit any stochastic pulsing! This does not seem to be analysed or discussed anywhere in the paper.

It is very odd for a supplementary figure like this to appear in a submitted manuscript. The only reference I could find to this simulation was in the methods “For the discrete simulations, we used Gillespie’s direct stochastic simulation algorithm” but I could not find any mention of the results in the main text. The potential difference between the two types of dynamics is potentially a major scientific issue with the current analysis. It is of utmost importance to clarify for which part (if any) of the results discrete Gillespie simulations were run/relevant. Note, the supplementary figure S1 did not specify any of the parameter values apart from pinit=0. (Incidentally, why was pinit=0 chosen here when pinit=0.001 was specified in the parameter table?) It also seems the Gillespie simulations were done in an unrealistic parameter regime for just a handful of phosphatase molecules. Taken at face value the plots would suggest that the authors’ main conclusions in the main text are misleading. This highlights a serious scientific issue with the current manuscript.

Overall presentation is exceedingly difficult to follow. The key assumptions and which type of model was used for which kind of result should be made much clearer. The actual numerical details of the methods are crucial but either demoted to the methods or not even mentioned. Some key aspects should be folded into the main results text so it is possible to understand what quantity of which model under which perturbation one is looking at.

For example, the distinction between single pulse and stochastic pulse dynamics is absolutely crucial for all their results. It should be explained as part of the results and not just in the methods. In fact, looking at the methods section it looks like the distinction is only made based on “time”, i.e., a single deterministic pulse is a response that is early, and everything that is late is classified as stochastic pulsing? If true, this should be highlighted. Also, the authors would need to show that this time-scale classification actually works to distinguish the two behaviours in *all* parameter regimes.

Note, that even after careful reading I do not fully understand the definitions of the responses. In line 328 the authors define the crucial measure as “we measure the magnitude of the single response pulse behaviour as the mean transient pulse amplitude divided by the mean asymptotic pulse amplitude”. However, the preceding bullet points do not define the “mean asymptotic pulse amplitude” but the "maximum asymptotic pulse amplitude" and "mean activity in the asymptotic state".

It is thus unclear what the term “mean asymptotic pulse amplitude” refers to. Imprecisions like this in language/presentations are extremely problematic and preclude the manuscripts’ results to contribute to the scientific literature.

Line 113 The authors claim "Through sample simulations, we demonstrated that the core circuit can generate both response behaviours, without making any assumptions about the upstream pathways (Figure 2D,E)." I disagree with the statement that no assumptions about the upstream pathways were made because the whole plot is for a *specific* type of upstream dynamics. I might be confused, but it doesn’t look like the authors meant what they stated here.

Line 101 “Tuning of a combination of parameters can also induce pulsing. However, we wished to minimise the number of model parameter values we modified. The main reason for this is that the original parameter set was picked to be biologically plausible [41], making minimal deviations from this set desired. In addition, modifying a smaller set of parameters reduces the dimensionality of the parameter space we need to explore, reducing computational complexity. We thus selected k_{K2} only as our proxy for the system’s proneness to pulsing.”

The argument to focus on just one parameter to avoid the complexity of the system’s behaviour seems like an argument based on convenience rather than science.

Making minimal deviations from the biologically plausible regime is sensible indeed, but that does not constrain the *number* of parameters changed. It constrains *how much* parameters should be changed. Having said that, I have no good intuition for how constrained the plausible range of the parameters presented in Table 2 really is. Are changes plus/minus a factor of two or a factor of 10 still plausible? I’d expect so, but the authors should clarify. For example, when the authors change k_{K2} the seem to deviate from the stated value of 36/h by a factor of 10. Why would it not be biologically plausible to change any of the other parameters similarly? Also, k_{K2} was changed only in one direction. Is it necessary for k_{K2} to be less than the plausible value?. I did not find a discussion of this anywhere.

Line 320 The authors state “To determine the magnitude of the behaviours for a given parameter set p… we simulate the system n times (n = 50, 100, or 200)" This is confusing. Which n is it?

There is a lack of detail and precision throughout the results figures. To pick out just a few that stood out the most to me:

FIG 2

Panel B has response the scale missing. What is the intuitive explanation for the increase in amplitude with noise when all other parameters are equal? That seems not intuitive at all.

The noise parameter is specified, but what are the other parameters? Are those the ones specified in Table 2? In particular, what was k_{K2} for the simulations in panel A,B?

Are the response amplitudes in panel A/B biologically realistic? If so, what about the response amplitudes in panels C/D which seem to be up to two orders of magnitude higher!

In Table 2 eta = 0.05 is specified. Where does that number come from, what does it mean intuitively, and is it biologically plausible?

Panel C/D is about the difference in single response pulse magnitude (SRPM) vs the stochastic pulsing amplitude (SPA) neither were defined at this point. There is not even a reference to the methods where the measures are defined. Only a reference to the supplementary figure S5. But this definition is key to everything that follows and cannot just be explained by a reference to a supplementary figure.

The result depicts the maximum SPA vs SRPM that can be achieved by changing p_stress. How exactly was this done, e.g., what is the range over which p_stress was varied? How did p_stress differ that maximizes the two different responses?

Panel E/F is described as depicting simulations that “maximise the pulses”. But it is unclear what parameters were varied to maximise the response. If I had to guess, specific eta and k_{K2} were chosen. What were those, and what is the p_stress that corresponded to those simulations?

What are the units of p_stress and k_{K2}? Presumably, they are micromole and 1/hour (as in Table 2) but in the main text the units seem to have been casually dropped throughout.

The conclusion of Fig 2 reads "This demonstrates that the CRN of the Narula model can generate both behaviours, when exposed to intrinsic noise only." I disagree that this model is one that is exposed to intrinsic noise only. Because of the global noise parameter that affects all intrinsic components it seems that some of the noise effect is due to what would be commonly called “extrinsic” sources.

FIG 3

Panel A/B What are the parameter values chosen, for example p_init, but also all others? The results here confused me because it seems like the model ever only leads to significant stress responses for very specific values of the product P_prod = kp*P_stress. Taken at face value that would suggest that a fine-tuning of kp to the stress-level would be necessary for the response module to function properly. That seems to imply that the model does not behave like the real system? Alternatively, it is possible that my interpretation here is due to the graphical scale chosen. In essence, everything here looks black which might possibly include a lot of responses like the ones depicted in Fig 2? The graphical presentation is thus problematic because it is not possible to infer the response behaviour.

What is the meaning of the parameter regime with significant response in SRPM and SPA? Looks like kp*P_stress~50 would lead to both responses. Which seems odd because the system would exhibit identical responses when all parameters are fixed. Does that mean the SRPM and SPA are non-exclusive or that the system randomly shows either behaviour? What are the implications for the interpretation of SRPM and SPA? What are experimental estimates for P_prod?

Panel C/D what are the values of the other parameters?

Panel E/F: The label suggests the plotting of a “cumulative distribution. I am not sure I understand how it is defined. Eq. 4 does not look like a cumulative distribution and the function plotted does not reach one as would be the case for a cumulative distribution function.

FIG 4

Panel B: This shows that no stable state exists for intermediate P_prod ranges. How was stability defined and determined in this stochastic system? All I could find is a reference to “BifurcationKit package (which tracks steady states using pseudo-arclength continuation)” which is not sufficient. What exactly is the P_prod range for unstable steady states? The lack of axis ticks makes it difficult to read, but eyeballing suggests that maybe around P_prod~125 the system is unstable. However, the simulation depicted in panel K of this figure looks like a stable system.

Fig 5

Panel A/B: What are the values of the other parameters? What is the distinctness -- how is it defined and what does it mean? At the very least the name of the variable that is plotted and where it is defined needs to be stated. I could not figure out what is plotted even after reading the methods section.

MINOR ISSUES

Line 88 Typo "noted"

Line 147 typo "we sat"

Multiple typos between line 266 and the equation "We will primarily be the" and "simualte", "described"

Line 269 "nosie"

Line 273 "acording"

Line 346 "ou" should be out

Line 348 typo "p_start"

Line 360 typo "if"

Typo page 18, caption of Fig 3 Ptress should be P_stress

Typo page 18, caption of Fig 3 "behaviorus" + grammar issue. Sentence needs to be fixed to make it is clear!

Fig 3 A/B show three lines but they are not the same but a different subset of the four values for kp*P_stress. It would be easier to compare the two panels if all four values of kp*P_stress were indicated in both panels.

Fig 3 E/F The legend indicates four parameters were changed but only three are visible. Maybe the fourth one is invisible because it is covered by another line?

Fig 5 caption. Typo "light green". Grammar in "For each parameter combination, the distinctness of the single response pulse (A) and stochastic pulsing (B) behaviours, as the parameter P_prod is varied (Methods)".

Fig 6 Missing scale for eta_{upstream}. And the notation is confusing as previously defined eta and eta_amp are relabelled as eta_core and eta_upstream.

Reviewer #3: The paper by Loman and Locke investigates a stochastic model of alternative sigma factor network for their ability to generate different types of pulsing dynamics as experimentally observed. As authors nicely summarise the state of the art, it is currently believed that the system produces alternative pulsing behaviour depending on the type of input it receives. However, the authors turn an existing ODE model of the system into a stochastic model using the chemical Langevin framework and show that the system indeed has the capability of exhibiting both behaviour in certain parameter regime. They further show that the interaction with the noise in the input can shape the tendency of the system to exhibit one type of behaviour vs another. Overall, the results nicely explain some recent unexpected experimental results in the system and suggests a series of new experiments. Also, more broadly, the study provides an example of how stochastic modelling of chemical networks can shed light on the type of behaviour they could exhibit. Overall, this is very interesting paper and fits well in the PLOS Comp Biol. It is nicely and clearly written. I particularly liked the fact that the main results section is short and to the point with details fully explained in the methods. I do not have any major comment. I include a few relatively minor points.

- The CLE framework brings in intrinsic noise in a particular way. In general, one could expect the noise in the different components of the system be a function of the parameters of the system, levels of mRNA and burstiness of the different genes. Could the authors provide a longer discussion of the limitations and the generality of the results obtained by the CLE.

- I have spotted a few typos in the paper, please read the paper another time carefully. For example on Page 8. “We will primarily be [use?] the CLE to simulate the model”

- A link to a github with code would be good. There is a vague statement in the paper about Code availability.

**Have the authors made all data and (if applicable) computational code underlying the findings in their manuscript fully available?**

Reviewer #1: Yes

Reviewer #2: Yes

Reviewer #3: Yes

PLOS authors have the option to publish the peer review history of their article (what does this mean?). If published, this will include your full peer review and any attached files.

Reviewer #1: No

Reviewer #2: No

Reviewer #3: No
---

## [Decision Letter · Decision Letter 1]

11 Apr 2023

Dear Dr. Locke,

Thank you very much for submitting your manuscript "The σ^B^ alternative sigma factor circuit modulates noise to generate different types of pulsing dynamics" for consideration at PLOS Computational Biology. As with all papers reviewed by the journal, your manuscript was reviewed by members of the editorial board and by several independent reviewers. The reviewers appreciated the attention to an important topic. Based on the reviews, we are likely to accept this manuscript for publication, providing that you modify the manuscript according to the review recommendations.

Most reviewers are satisfied with the revisions and it is likely that the manuscript will be accepted. However, reviewer 2 still has some concerns. When you finalize the manuscript, please address these as far as possible and reply to them point-by-point.

Sincerely,

Tobias Bollenbach

Academic Editor

PLOS Computational Biology

Mark Alber

Section Editor

PLOS Computational Biology

Most reviewers are satisfied with the revisions and it is likely that the manuscript will be accepted. However, reviewer 2 still has some concerns. When you finalize the manuscript, please address these as far as possible and reply to them point-by-point.

Reviewer's Responses to Questions

**Comments to the Authors:**

Reviewer #1: The authors have carefully revised the manuscript and answered all my questions. I am happy to recommend publication.

Reviewer #2: The authors made extensive changes that have significantly improved the manuscript. Furthermore, the authors' replies have clarified some misunderstandings that I had about their work. However, even the revised and improved version has some significant issues that should to be further improved.

For a start, the authors should heed the previously expressed advice of referee #3 to "please read the paper another time carefully". This still applies even after this round of revisions. Just to highlight two egregious examples: 1) the supplemental file currently has all references to the bibliography and cross-references replaced by question marks. 2) the rewritten key definitions of the pulse amplitudes now contain mistakes that mix up the transient and asymptotic phases. For example, "The maximum asymptotic pulse amplitude... is measured as the maximum activity of the simulation in the transient phase." The peer review process should not be a substitute for careful proofreading and pre-submission quality control.

The main focus of my review are the used methods. I have no issue with the CLE now that the authors have shown that Gillespie simulations exhibit the same behaviour. But I think the authors would be well advised to clarify more what their "behavioural measures" are and what they mean. The authors' reply clarified that I incorrectly thought that in Fig. 2 the "Single response pulse magnitude" referred to the same thing as the "amplitude of the single response pulse". I suspect I will not be the only reader confused by the terminology because the terms amplitude and magnitudes are often used interchangeably. (For example, in this manuscript the authors themselves use noise magnitude and noise amplitude interchangeably.) It is thus problematic to redefine a pulse amplitude into a pulse magnitude by rescaling it by another amplitude. This is not a "mere issue of terminology" but is about an important scientific ingredient that is crucial to understand all the authors' results, namely what exactly the "Single response pulse magnitude" and the ""Stochastic response pulse magnitude" measure. The authors would be well advised to illustrate this key definition in the main text. One way would be, e.g., to promote the supplemental figure SFig. 4 to the main text.

The problem of defining response magnitudes that are not directly amplitudes is compounded by the fact that much of subsequent results are for quantities further derived from these magnitudes, e.g., from Msrp and Msp the authors define Msrp* as max(0, Msrp-Msp). The authors should explicitly demonstrate that such derived measures are meaningful. For example, is a system with Msrp*=0 because because Msrp=5 and Msp=7 really a much more stochastic response then say a Msrp=12 and Msp=10 system? This is relevant because the "phase space" analysis rests on quantities further derived from Mrsp* depends on parameters.

To me it seems the key scientific contribution of this manuscript lies in determining a "phase space" of behaviour. In particular, to quantify how the system behaves as a function of the p_prod. But such an analysis strictly relies on the reported quantities to be meaningful. For example, one thing to discuss more explicitly is how these measures characterize/quantify oscillatory systems. When the system oscillates I'd expect for the stochastic pulsing magnitude (SPM) to be around 2 or 3. I suspect the authors don't deem that a relevant magnitude but they should argue that explicitly. Similarly, the SPM seems to be maximal when there is exactly one stochastic pulse in the time-interval [5,200]. And as soon as a second or third one appears the SPM measure severely drops. It would be good to describe/comment on this.

Regarding the single response pulse magnitude I raised a technical question in the last review that I am not sure is entirely addressed by the authors' response. Their pulse distinction is based on an early/transient time interval [0,5] and a late asymptotic one. But that only makes sense if there aren't any stochastic pulses within [0,5], i.e. that the early interval only ever contains either 0 or 1 but never 2 pulses. Could the authors verify that this is indeed the case? The current statement is a bit unclear on this at it seems to talk about the length of the initial pulse, which indeed seem much shorter than that interval. "Finally, to ensure that the transient phase indeed never surpasses 5 hours, we made an automatic scan of a large number (10, 000) of random parameter sets of our model. None of these exhibited an initial pulse longer than 5 hours."

To me it seems that the authors' main finding is that in their modelled system noise can induce stochastic pulsing in a regime before the system becomes oscillatory. Such behaviour is interesting and important. But the authors could embed their findings in the existing literature on such noise induced oscillations and stochastic pulses when a stability analysis of the deterministic system would suggest a single stable fixed point [e.g. their results and how they relate to references 55,56,57 could be discussed in the discussion]. It also seems to be that the stability analysis of the deterministic Narula model is novel and interesting. In the revised manuscript it has been demoted to the supplement. [Possibly in response to one of my comments about the difference between the unstable region of the deterministic system vs the stochastic one. But instead of hiding this difference in the supplement I think it would be much more interesting/appropriate to discuss it explicitly!]

I still found the presentation of the model confusing because it is done in two steps separated over two section 4.1 and 4.2. That leads to seemingly contradictory statements because in both sections the authors refer to 'the model' and 'the parameters' but the meaning changes between the sections. Would it make more sense to just define the final model the authors analyze? (This would also avoid the repetitive ODE table that is mostly overlap between 4.1 and 4.2.) The authors should of course still explicitly discuss the novel parts introduced in their model compared to the published Narula model. But I think doing so by formally defining two slightly different models is counterproductive.

Some technical points that should be fixed.

The authors state "to prevent negative numbers in our CLE simulations we use a reflective boundary condition". However, when they describe their numerical method it did not sound like they actually implemented reflective boundary condition, but that their code allows for negative numbers and they employed a 'hack' to deal with the subsequent negative reaction rates by taking absolute values of the rates in the stochastic term. "The CLE reflective boundary condition is implemented by, for every term within a square root, replacing it with its absolute value."

Below is a (probably incomplete) list of typos that I noticed throughout the manuscript. I implore the authors to take more care in their preparation of future revisions. Peer review should not be used as a substitute for proofreading.

Author summary

Typo: bet-bedge

Main text typos

Line 117 "We note that the CLE and Gillespie algorithm both models intrinsic noise only". models -> model

Line 296 "Ornstein-Uhlenbreck process " Uhlenbreck -> Uhlenbeck

L385 magntiude

L447, 450 Fix quotation marks.

L454 implemenpted

L458 pushes

L477 "The amplitude of the transient pulse. This is measured as the maximum activity of the simulation in the assymptotic phase (t in (0.0, 5.0))."

I suspect the authors mean transient rather than assymptotic [sic] here. Similarly, in the next definition: "The mean asymptotic activity. This is measured as the mean activity of the simulation in the transient phase (t in [5.0, 200.0])." I suspect the authors mean asymptotic rather than transient here.

L524 change -> changes

Fig 4 caption "A similar transition, but generated through Gillespie algorithm simulations, can be found in Supplementary figure 19"

I suspect the authors mean Supplementary Fig 18.

Fig 4 caption "12 different selected values of pprod (used in C-N)"

I think the authors mean B-M in the current iteration of this figure.

ODE systems: in first line v0 should be v_0

In general, the authors correctly follow the convention of separating numbers and units with a space and non-italicized units. But sometimes they don't. The presentation should be made consistent.

Supplement

"subsitutions "

SFig 4 caption typo "asymtotic" should be asymptotic, "the the" should be the.

SFIg 5. "upstreams" should be upstream

SFIg 2 axis label typo "stady" should be steady

SFig 7 caption "this examples"

All references and cross-references currently are ? or ??.

Reviewer #3: The authors have fully revised the paper addressing my concerns.

**Have the authors made all data and (if applicable) computational code underlying the findings in their manuscript fully available?**

Reviewer #1: Yes

Reviewer #2: Yes

Reviewer #3: Yes

PLOS authors have the option to publish the peer review history of their article (what does this mean?). If published, this will include your full peer review and any attached files.

Reviewer #1: No

Reviewer #2: No

Reviewer #3: **Yes: **Vahid Shahrezaei

Figure Files:

Data Requirements:

Reproducibility:

References:

---

## [Editor Report · Decision Letter 2]

12 Jun 2023

Dear Dr. Locke,

We are pleased to inform you that your manuscript 'The σ^B^ alternative sigma factor circuit modulates noise to generate different types of pulsing dynamics' has been provisionally accepted for publication in PLOS Computational Biology.

Best regards,

Tobias Bollenbach

Academic Editor

PLOS Computational Biology

Mark Alber

Section Editor

PLOS Computational Biology

---

## [Editor Report · Acceptance letter]

28 Jul 2023

PCOMPBIOL-D-22-01479R2 

The σ^B^ alternative sigma factor circuit modulates noise to generate different types of pulsing dynamics

Dear Dr Locke,

I am pleased to inform you that your manuscript has been formally accepted for publication in PLOS Computational Biology. Your manuscript is now with our production department and you will be notified of the publication date in due course.

With kind regards,

Timea Kemeri-Szekernyes
